# Transformer with a Mixture of Gaussian Keys

## Abstract

Multi-head attention is a driving force behind state-of-the-art transformers which achieve remarkable performance across a variety of natural language processing (NLP) and computer vision tasks. It has been observed that for many applications, those attention heads learn redundant embedding, and most of them can be removed without degrading the performance of the model. Inspired by this observation, we propose Transformer with a Mixture of Gaussian Keys (Transformer-MGK), a novel transformer architecture that replaces redundant heads in transformers with a mixture of keys at each head. These mixtures of keys follow a Gaussian mixture model and allow each attention head to focus on different parts of the input sequence efficiently. Compared to its conventional transformer counterpart, Transformer-MGK accelerates training and inference, has fewer parameters, and requires less FLOPs to compute while achieving comparable or better accuracy across tasks. Transformer-MGK can also be easily extended to use with linear attentions. We empirically demonstrate the advantage of Transformer-MGK in a range of practical applications including language modeling and tasks that involve very long sequences. On the Wikitext-103 and Long Range Arena benchmark, Transformer-MGKs with 4 heads attain comparable or better performance to the baseline transformers with 8 heads.

## 1 Introduction

Transformers (Vaswani et al., 2017) have become the state-of-the-art model for sequence processing tasks, solving many challenging problems in natural language processing and computer vision (Al-Rfou et al., 2019; Dai et al., 2019; Williams et al., 2018; Devlin et al., 2018; Brown & et al., 2020; Howard & Ruder, 2018; Rajpurkar et al., 2016; Dehghani et al., 2018; So et al., 2019; Dosovitskiy et al., 2020; Touvron et al., 2020). These models can also transfer the learned knowledge from a pre-trained model to task that involves different data modalities and has limited supervision (Radford et al., 2018; 2019; Devlin et al., 2018; Yang et al., 2019; Liu et al., 2019). The success of transformers is rooted in the self-attention mechanism as their fundamental building blocks for modeling (Cho et al., 2014; Parikh et al., 2016; Lin et al., 2017). For each token, self-attention computes a weighted average of the feature representations of other tokens where the weight is proportional to a similarity score between each pair of tokens. This mechanism allows a token to pay attention to other tokens in the sequence and attain a contextual representation (Bahdanau et al., 2014; Vaswani et al., 2017; Kim et al., 2017). It has been shown that the representation capacity of the attention mechanism (Tenney et al., 2019) and its capability of capturing diverse syntactic and semantic relationships (Tenney et al., 2019; Vig & Belinkov, 2019; Clark et al., 2019; Voita et al., 2019a; Hewitt & Liang, 2019) is key to the impressive performance of transformers in practice.

### 1.1 Self-Attention

For a given input sequence $\mathbf{X} := [\boldsymbol{x}_1, \cdots, \boldsymbol{x}_N]^\top \in \mathbb{R}^{N \times D_x}$ of $N$ feature vectors, self-attention transforms $\mathbf{X}$ into the output sequence $\mathbf{H}$ in the following two steps:

Step 1. The input sequence $\boldsymbol{X}$ is projected into the query matrix $\mathbf{Q}$, the key matrix $\mathbf{K}$, and the value matrix $\mathbf{V}$ via three linear transformations

$$\mathbf{Q} = \mathbf{X}\mathbf{W}_Q^\top; \mathbf{K} = \mathbf{X}\mathbf{W}_K^\top; \mathbf{V} = \mathbf{X}\mathbf{W}_V^\top,$$

where $\mathbf{W}_Q, \mathbf{W}_K \in \mathbb{R}^{D \times D_x}$, and $\mathbf{W}_V \in \mathbb{R}^{D_v \times D_x}$ are the weight matrices. We denote $\boldsymbol{Q} := [\boldsymbol{q}_1, \cdots, \boldsymbol{q}_N]^\top, \mathbf{K} := [\boldsymbol{k}_1, \cdots, \boldsymbol{k}_N]^\top$, and $\mathbf{V} := [\boldsymbol{v}_1, \cdots, \boldsymbol{v}_N]^\top$, where the vectors $\boldsymbol{q}_i, \boldsymbol{k}_i, \boldsymbol{v}_i$ for $i = 1, \cdots, N$ are the query, key, and value vectors, respectively.

Step 2. The output sequence $\mathbf{H} := [\boldsymbol{h}_1, \cdots, \boldsymbol{h}_N]^\top$ is then computed as follows

$$\mathbf{H} = \text{softmax}\Big(\mathbf{Q}\mathbf{K}^\top / \sqrt{D}\Big)\mathbf{V} := \mathbf{A}\mathbf{V}, \tag{1}$$

where the softmax function is applied to each row of the matrix $(\mathbf{Q}\mathbf{K}^\top)/\sqrt{D}$. For each query vector $\boldsymbol{q}_i$ for $i = 1, \cdots, N$, an equivalent form of Eqn. (1) to compute the output vector $\boldsymbol{h}_i$ is given by

$$\boldsymbol{h}_i = \sum_{j=1}^{N} \text{softmax}\Big(\boldsymbol{q}_i^\top \boldsymbol{k}_j / \sqrt{D}\Big)\boldsymbol{v}_j := \sum_{j=1}^{N} a_{ij}\boldsymbol{v}_j. \tag{2}$$

The matrix $\mathbf{A} \in \mathbb{R}^{N \times N}$ and its component $a_{ij}$ for $i, j = 1, \cdots, N$ are the attention matrix and attention scores, respectively. The self-attention computed by Eqn. (1) and (2) is called the scaled dot-product attention or softmax attention. In our paper, we call a transformer that uses this attention the softmax transformer. The structure that the attention matrix $\mathbf{A}$ learns from training determines the ability of the self-attention to capture contextual representation for each token.

**Multi-head Attention** Each output sequence $\mathbf{H}$ forms an attention head. In multi-head attention, multiple heads are concatenated to compute the final output. Let $H$ be the number of heads and $\mathbf{W}^O \in \mathbb{R}^{HD_v \times HD_v}$ be the projection matrix for the output. The multi-head attention is defined as

$$\text{MultiHead}(\{\mathbf{Q}\}_{i=1}^H, \{\mathbf{K}\}_{i=1}^H, \{\mathbf{V}\}_{i=1}^H) = \text{Concat}(\mathbf{H}_1, \mathbf{H}_2, \ldots, \mathbf{H}_H)\mathbf{W}^O. \tag{3}$$

Even though multi-head attention extends single-head attention to capture diverse attention patterns and improve the performance of transformers, it has been shown that transformers for practical tasks including sequence classification and language modeling learn redundant heads (Michel et al., 2019). These redundant heads compute similar attention mappings. Having many of them in the model limits the representation capacity of the transformer while wasting parameters, memory and computation, impeding the application of transformers to many important large-scale tasks.

## 1.2 CONTRIBUTION

We establish the correspondence between self-attention in transformer and a Gaussian mixture model (GMM) and propose Transformer with a Mixture of Gaussian Keys (Transformer-MGK), a novel class of transformers that can avoid the head redundancy. At the core of Transformer-MGK is replacing the attention key $\boldsymbol{k}_j$ in each head by a GMM to allow the query $\boldsymbol{q}_i$, as well as its associated token, to attend to more diverse positions in the input sequence, thereby increasing the representation of each attention head and reducing the chance of learning redundant heads. Our contribution is four-fold:

1. We construct a GMM and show that attention scores in self-attention match posterior distribution in our model, providing a probabilistic framework to study self-attention in transformers.

2. Under our probabilistic framework for self-attention, we introduce an additional mixture of Gaussian to model each attention key. We empirically show that this mixture of Gaussian keys (MGK) can capture a diversity of attention patterns, thus alleviating head redundancy.

3. We extend our MGK to use with linear attentions and propose the mixture of linear keys (MLK) for efficient computation and better memory footprint.

4. We empirically show that Transformer-MGK and Transformer-MLK are comparable or better than the corresponding baseline transformers with softmax and linear attentions while only using half the number of attention heads and reducing both model complexity measured by the number of parameters and computational cost in terms of FLOPs.

**Organization:** We structure this paper as follows: In Section 2, we establish the connection between GMM and self-attention and then present our Transformer-MGK and its extensions including Transformer-MLK. In Section 3, we validate and empirically analyze the efficiency and accuracy of Transformer-MGK/MLK. We discuss related works in Section 4. The paper ends up with concluding remarks. More experimental details are provided in the Appendix.

## 2 TRANSFORMER WITH A MIXTURE OF GAUSSIAN KEYS

### 2.1 ATTENTION SCORE AS A POSTERIOR DISTRIBUTION

We first consider a query $\boldsymbol{q}_i \in \mathbf{Q}$ and a key $\boldsymbol{k}_j \in \mathbf{K}$. Let $\boldsymbol{t}$ be a $N$-dimensional binary random variable having a 1-of-$N$ representation in which a particular element $\boldsymbol{t}_j$ is equal to 1 and all other

elements are equal to 0. We use $t_j$ to indicate the position $j$ of the key $k_j$. In particular, let $\mathbf{I}$ be the identity matrix, we model the distribution $p(q_i)$ by the following GMM:

$$p(\boldsymbol{q}_i) = \sum_{j=1}^{N} \pi_j p(\boldsymbol{q}_i | \boldsymbol{t}_j = 1) = \sum_{j=1}^{N} \pi_j \mathcal{N}(\boldsymbol{q}_i \,|\, \boldsymbol{k}_j, \sigma_j^2 \mathbf{I}), \qquad (4)$$

where $\pi_j$ is the prior $p(\boldsymbol{t}_j = 1)$. Given the query $\boldsymbol{q}_i$, how likely $\boldsymbol{q}_i$ matches the key $\boldsymbol{k}_j$ is given by posterior $p(\boldsymbol{t}_j = 1 | \boldsymbol{q}_i)$. This posterior is computed as follows

$$
\begin{aligned}
p(\boldsymbol{t}_j = 1 | \boldsymbol{q}_i) &= \frac{\pi_j \mathcal{N}(\boldsymbol{q}_i \,|\, \boldsymbol{k}_j, \sigma_j^2)}{\sum_{j'} \pi_{j'} \mathcal{N}(\boldsymbol{q}_i \,|\, \boldsymbol{k}_{j'}, \sigma_{j'}^2)} = \frac{\pi_j \exp\left(-\|\boldsymbol{q}_i - \boldsymbol{k}_j\|^2 / 2\sigma_j^2\right)}{\sum_{j'} \pi_{j'} \exp\left(-\|\boldsymbol{q}_i - \boldsymbol{k}_{j'}\|^2 / 2\sigma_{j'}^2\right)} \\
&= \frac{\pi_j \exp\left[-\left(\|\boldsymbol{q}_i\|^2 + \|\boldsymbol{k}_j\|^2\right) / 2\sigma_j^2\right] \exp\left(\boldsymbol{q}_i \boldsymbol{k}_j^\top / \sigma_j^2\right)}{\sum_{j'} \pi_{j'} \exp\left[-\left(\|\boldsymbol{q}_i\|^2 + \|\boldsymbol{k}_{j'}\|^2\right) / 2\sigma_{j'}^2\right] \exp\left(\boldsymbol{q}_i \boldsymbol{k}_{j'}^\top / \sigma_{j'}^2\right)}.
\end{aligned}
\qquad (5)
$$

We further assume that the query $\boldsymbol{q}_i$ and the key $\boldsymbol{k}_j$ are normalized, and the prior $\pi_j$ is uniform. We will justify these assumptions in our Remarks at the end of this section. We also let $\sigma_j^2 = \sigma^2$, $j = 1, 2, \ldots, K$. Then the posterior $p(\boldsymbol{t}_j = 1 | \boldsymbol{q}_i)$ can be written as

$$p(\boldsymbol{t}_j = 1 | \boldsymbol{q}_i) = \exp\left(\boldsymbol{q}_i \boldsymbol{k}_j^\top / \sigma^2\right) / \sum_{j'} \exp\left(\boldsymbol{q}_i \boldsymbol{k}_{j'}^\top / \sigma^2\right). \qquad (6)$$

The right-hand side of Eqn. (6) matches the attention score given in Eqn. (2) when $\sigma^2 = \sqrt{D}$. Thus, we show that under right assumptions, the attention score between the query $\boldsymbol{q}_i$ and the key $\boldsymbol{k}_j$ in an attention unit of a transformer is the posterior $p(\boldsymbol{t}_j = 1 | \boldsymbol{q}_i)$, which indicates the *responsibility* that the key $\boldsymbol{k}_j$ takes for 'explaining' the query $\boldsymbol{q}_i$, which in turn decide, for example, how much a token at position $i$ pays attention to a token at position $j$ in the input sequence.

**Remark 1** *The assumption that the query $\boldsymbol{q}_i$ and the key $\boldsymbol{k}_j$ are normalized is realistic and not artificial. In many applications, those two vectors are normalized. Schlag et al. (2021) points out that such normalization is to avoid instability occurring during the training.*

**Remark 2** *In practice, the prior is chosen to be uniform when there is no prior knowledge available.*

## 2.2 Transformer with a Mixture of Gaussian Keys: Each Key is Again a Gaussian Mixture Model

As we have seen from Eqn. (6), the key $k_j$ is used to explain the query $q_i$ via the posterior $\mathbb{P}(t_j = 1 | q_i)$. Via this simple connection, each query $q_i$ is treated to be as a sample from the mixture of $N$ keys $\sum_{j=1}^{N} \pi_j \mathcal{N}(q_i | k_j, \sigma_j^2 I_d)$. However, the assumption that the distribution $\mathbb{P}(q_i | t_j = 1)$ at each subpopulation is Gaussian in Eqn. (4) can be quite strong as there is no guarantee that this assumption is valid in practice. In particular, it may happen that the distribution of each subpopulation is asymmetric or skewed or even multimodal. Therefore, using the Gaussian distribution for each subpopulation can potentially limit the explanation power and diversity of each subpopulation/ key. It indicates that we should use more expressive distributions to represent $\mathbb{P}(q_i | t_j = 1)$.

**Mixture of Gaussian keys:** To improve the explanation power of each key $k_j$, potentially increase the representation of each attention head, and reduce the chance of learning redundant heads, we would like to model it as mixture of Gaussian distributions. We refer to this model as *Transformer with a Mixture of Gaussian Keys* (Transformer-MGK). In particular, in Transformer-MGK we model each key $k_j$ at position $j$ as a mixture of $M$ Gaussians $\mathcal{N}(k_{jr}, \sigma_{jr}^2 \mathbf{I})$, $r = 1, 2, \ldots, M$. Here we are overloading the notation a little bit and use $k_{jr}$ and $\sigma_{jr}^2 \mathbf{I}$ to denote the mean and covariance matrix of the $r^{th}$ Gaussian at position $j$. Let $z$ be a $M$-dimensional binary random variable having a 1-of-$M$ representation. We use $z_r$ to indicate the $r^{th}$ Gaussian in the mixture. Let $\pi_{jr} \equiv \mathbb{P}(z_r = 1 | t_j = 1)$, our MGK can be written as

$$\mathbb{P}(\boldsymbol{q}_i | \boldsymbol{t}_j = 1) = \sum_r \mathbb{P}(\boldsymbol{z}_r = 1 | \boldsymbol{t}_j = 1) \mathbb{P}(\boldsymbol{q}_i | \boldsymbol{z}_r = 1, \boldsymbol{t}_j = 1) = \sum_r \pi_{jr} \mathcal{N}(\boldsymbol{q}_i \,|\, \boldsymbol{k}_{jr}, \sigma_{jr}^2 \mathbf{I}). \qquad (7)$$

Our motivation of using mixture of Gaussian distributions to represent the distribution of each subpopulation in Eqn. (7) stems from the following important approximation result:

**Theorem 1** *Assume that $P$ is probability distribution on $[-a, a]^d$ for some $a > 0$ and admits density function $p$ such that $p$ is differentiable and bounded. Then, for any given variance $\sigma > 0$ and for any $\epsilon > 0$, there exists a mixture of $K$ components $\sum_{i=1}^{K} \pi_i \mathcal{N}(\theta_i, \sigma^2 \mathbf{I})$ where $K \leq (C \log(1/\epsilon))^d$ for some universal constant $C$ such that*

$$\sup_{x \in \mathbb{R}^d} |p(x) - \sum_{i=1}^{K} \pi_i \phi(x|\theta_i, \sigma^2 \mathbf{I})| \leq \epsilon,$$

*where $\phi(x|\theta, \sigma^2 \mathbf{I})$ is the density function of multivariate Gaussian distribution with mean $\theta$ and covariance matrix $\sigma^2 \mathbf{I}$.*

The proof of Theorem 1 is in Appendix C. The result of Theorem 1 suggests that regardless of the real form of $\mathbb{P}(\boldsymbol{q}_i | \boldsymbol{t}_j = 1)$, we can use finite mixture of Gaussian distributions to approximate $\mathbb{P}(\boldsymbol{q}_i | \boldsymbol{t}_j = 1)$. It allows us to have richer approximation of $\mathbb{P}(\boldsymbol{q}_i | \boldsymbol{t}_j = 1)$ than by using a simple Gaussian distribution in Eqn. (4). Similar to the derivation above, the posterior $p(\boldsymbol{t}_j = 1 | \boldsymbol{q}_i)$ in Transformer-MGK can be written as

$$\mathbb{P}(\boldsymbol{t}_j = 1 | \boldsymbol{q}_i) = \frac{\sum_r \pi_{jr} \exp\left(\boldsymbol{q}_i \boldsymbol{k}_{jr}^{\top} / \sigma_{jr}^2\right)}{\sum_{j'} \sum_r \pi_{j'r} \exp\left(\boldsymbol{q}_i \boldsymbol{k}_{j'r}^{\top} / \sigma_{j'r}^2\right)}. \tag{8}$$

Furthermore, in Transformer-MGK, we relax the assumption that the queries and keys are normalized. Thus, when computing $\mathbb{P}(\boldsymbol{t}_j = 1 | \boldsymbol{q}_i)$, we compute the Gaussian kernels between the queries and keys instead of their dot products. The posterior $\mathbb{P}(\boldsymbol{t}_j = 1 | \boldsymbol{q}_i)$ in Transformer-MGK is then given by

$$\mathbb{P}(\boldsymbol{t}_j = 1 | \boldsymbol{q}_i) = \frac{\sum_r \pi_{jr} \exp\left(-\|\boldsymbol{q}_i - \boldsymbol{k}_{jr}\|^2 / 2\sigma_{jr}^2\right)}{\sum_{j'} \sum_r \pi_{j'r} \exp\left(-\|\boldsymbol{q}_i - \boldsymbol{k}_{j'r}\|^2 / 2\sigma_{j'r}^2\right)}. \tag{9}$$

As proven in Section 2.1, this posterior corresponds to the attention score. Thus, Eqn. (9) is the formula for computing the attention score in Transformer-MGK. We compute the output vector $\boldsymbol{h}_i$ of the self-attention in Transformer-MGK as follows

$$\boldsymbol{h}_i = \sum_j \left( \frac{\sum_r \pi_{jr} \exp\left(-\|\boldsymbol{q}_i - \boldsymbol{k}_{jr}\|^2 / 2\sigma_{jr}^2\right)}{\sum_{j'} \sum_r \pi_{j'r} \exp\left(-\|\boldsymbol{q}_i - \boldsymbol{k}_{j'r}\|^2 / 2\sigma_{j'r}^2\right)} \right) \boldsymbol{v}_j. \tag{10}$$

## 2.3 INFERENCE AND LEARNING VIA THE EXPECTATION MAXIMIZATION ALGORITHM

Let $\gamma_{ir} \equiv \mathbb{P}(\boldsymbol{z}_r = 1 | \boldsymbol{q}_i, \boldsymbol{t}_j = 1)$, in MGK, we apply the E-step inference in the Expectation-Maximization (EM) algorithm to estimate this posterior given the query $\boldsymbol{q}_i$. The posterior $\gamma_{ir}$ is also known as the *responsibility* that the component $\mathcal{N}(\boldsymbol{k}_{jr}, \sigma_{jr}^2 \mathbf{I})$ takes to account for the observation, which in MGK is the query $\boldsymbol{q}_i$. Below we propose two approaches to estimate this responsibility.

**Soft E-step** Using soft E-step inference, the EM algorithm makes a soft assignment, in which each query is associated with all clusters. The responsibilities are then given by

$$\gamma_{ir} = \frac{\pi_{jr} \exp\left(-\|\boldsymbol{q}_i - \boldsymbol{k}_{jr}\|^2 / 2\sigma_{jr}^2\right)}{\sum_{r'} \pi_{jr'} \exp\left(-\|\boldsymbol{q}_i - \boldsymbol{k}_{jr'}\|^2 / 2\sigma_{jr'}^2\right)}. \tag{11}$$

At learning time, the responsibilities estimated by Eqn. (11) are used to update the prior $\pi_{jr}$, i.e. $\pi_{jr} = N_{jr}/N$, where $N$ is the number of queries and $N_{jr} = \sum_{i=1}^{N} \gamma_{ir}$. These updated priors $\pi_{jr}$ are then used in Eqn. (9) to compute attention scores.

**Hard E-step** Hard E-step performs a hard assignment of queries to key clusters, in which each query is associated uniquely with one cluster. This is similar to the $K$-means algorithm (Lloyd, 1982) and corresponds to the MGK at the limit when the variance parameter $\sigma_{jr}^2$ goes to 0. Following the derivation of $K$-means from a GMM in (Bishop, 2006), Eqn. (9) becomes

$$\mathbb{P}(\boldsymbol{t}_j = 1 | \boldsymbol{q}_i) = \frac{\max_r \exp\left(-\|\boldsymbol{q}_i - \boldsymbol{k}_{jr}\|^2 / 2\sigma_{jr}^2\right)}{\sum_{j'} \max_r \exp\left(-\|\boldsymbol{q}_i - \boldsymbol{k}_{j'r}\|^2 / 2\sigma_{j'r}^2\right)}. \tag{12}$$

**Remark 3** *The hard E-step inference allows the attention score to be computed more efficiently because the priors $\pi_{jr}$ no longer play an active role in the algorithm and can be completely ignored.*

**Learning via Stochastic Gradient Descent (SGD)** In order to increase the efficiency of the model, in MGK, we fix the variance parameter $\sigma_{jr}^2$ to be $\sqrt{D}$ as in the standard softmax attention and make the cluster means, i.e. the keys, $\boldsymbol{k}_{jr}$ learnable parameters. We also make the prior $\pi_{jr}$ learnable parameters as one of the design options. In that case, both $\boldsymbol{k}_{jr}$ and $\pi_{jr}$ are learned via SGD. This update via SGD can be considered as a generalized M-step (Bishop, 2006).

**Design Options for Keys** (Option A) We follow the standard setting in the softmax transformer and make the keys $\boldsymbol{k}_{jr}$ a linear projection of the input $\boldsymbol{x}_j$, i.e. $\boldsymbol{k}_{jr} = \boldsymbol{x}_j \mathbf{W}_{K_r}^\top$, where $\boldsymbol{x}_j \in \mathbb{R}^{1 \times D_x}$, $\mathbf{W}_{K_r} \in \mathbb{R}^{D \times D_x}$ and $r = 1, 2, \ldots, M$. (Option B) Alternatively, we also make the keys $\boldsymbol{k}_{jr}$ shifted version of each other to save computation, i.e. $\boldsymbol{k}_{jr} = \boldsymbol{x}_j \mathbf{W}_K^\top + \boldsymbol{b}_r$, where $\mathbf{W}_K \in \mathbb{R}^{D \times D_x}$.

## 2.4 TRANSFORMER WITH A MIXTURE OF LINEAR KEYS

The MGK can be easily extended to use with linear attentions. We call that model Transformer with a Mixture of Linear Keys (Transformer-MLK). In this section, we adopt the formulation of linear attentions from (Katharopoulos et al., 2020) to derive Transformer-MLK. Similar approach can be taken to derive Transformer-MLK when using with other linear attentions such as those in performers (Choromanski et al., 2021) and fast-weight transformers (Schlag et al., 2021). In Transformer-MLK, the Gaussian kernel in Eqn. (10) is linearized as the product of feature maps $\phi(\cdot)$ on the vectors $\boldsymbol{q}_i$ and $\boldsymbol{k}_j$. The associative property of matrix multiplication is then utilized to derive the following efficient computation of the attention map

$$\boldsymbol{h}_i = \frac{\sum_j \sum_r \pi_{jr} \phi(\boldsymbol{q}_i)^\top \phi(\boldsymbol{k}_{jr}) \boldsymbol{v}_j}{\sum_j \sum_r \pi_{jr} \phi(\boldsymbol{q}_i)^\top \phi(\boldsymbol{k}_{jr})} = \frac{\phi(\boldsymbol{q}_i)^\top \sum_j \sum_r \pi_{jr} \phi(\boldsymbol{k}_{jr}) \boldsymbol{v}_j^\top}{\phi(\boldsymbol{q}_i)^\top \sum_j \sum_r \pi_{jr} \phi(\boldsymbol{k}_{jr})}. \tag{13}$$

Replacing $\sum_j \sum_r \pi_{jr} \phi(\boldsymbol{q}_i)^\top \phi(\boldsymbol{k}_{jr}) \boldsymbol{v}_j$ with $\phi(\boldsymbol{q}_i)^\top \sum_j \sum_r \pi_{jr} \phi(\boldsymbol{k}_{jr}) \boldsymbol{v}_j^\top$, as in linear transformers, reduces the memory and computational cost of computing the attention map in Transformer-MLK from $\mathcal{O}(N^2)$ to $\mathcal{O}(N)$, making Transformer-MLK scalable to very long sequences.

## 3 EXPERIMENTAL RESULTS

In this section, we numerically justify the efficiency of Transformer-MGK/MLK and empirically study the advantage of using mixture of keys on various benchmarks, including different tasks in the Long Range Arena (LRA) (Tay et al., 2021) (Section 3.1) and language modeling on Wikitext-103 (Merity et al., 2017) (Section 3.2). We aim to show that: (i) Transformer-MGK/MLK with half the number of heads is comparable or better than the baseline softmax and linear transformers with the full number of heads while being more efficient in both computational cost and memory footprints; (ii) Mixture of keys helps reduce the redundancy in multi-head transformers and benefits learning of the long-term dependency in long input sequences; (iii) Using the same number of heads, Transformer-MGK/MLK significantly outperforms the baseline softmax and linear transformers. Especially in the case of Transformer-MLK, it helps reduce the performance gap between softmax and linear transformers.

Throughout this section, we compare Transformer-MGK/MLK with the softmax and linear transformers that have the same or double the number of attention heads. In all experiments, for our Transformer-MGK/MLK models, we set M=2 where M is the number of Gaussians, i.e. keys, at each timestep. Among the design options for Transformer-MGK mentioned in Section 2.3, we use the one with Soft-E step but make the parameter $\pi_{jr}$ and $\boldsymbol{k}_{jr}$ learnable and fix the variance $\sigma_{jr}^2$ to be constants. We study both implementations for keys: (A) $\boldsymbol{k}_{jr}$ is a linear projection of the input $\boldsymbol{x}_j$, i.e., $\boldsymbol{k}_{jr} = \boldsymbol{x}_j \mathbf{W}_{K_r}^\top$ and (B) $\boldsymbol{k}_{jr}$ are shifted version of each other, i.e., $\boldsymbol{k}_{jr} = \boldsymbol{x}_j \mathbf{W}_K^\top + \boldsymbol{b}_r$.

In this section, we refer to the Transformer-MGK/MLK whose keys are implemented by (A) as Transformer-MGK/MLK, and whose keys are implemented by (B) as Transformer-sMGK/sMLK. We empirically compare these models with other design options for Transformer-MGK in Section 3.4. Details on datasets, models, and training are provided in Appendix A.1.

## 3.1 LONG RANGE ARENA (LRA) BENCHMARK

**Models and baselines** We compare our 1-head, 2-head, 4-head Transformer-MGK and MLK with the baseline softmax (Vaswani et al., 2017) and linear transformers (Katharopoulos et al., 2020) that have 1 head, 2 heads, 4 heads, and 8 heads. Each model consists of two layers, and we adopt the model and training setting from (Xiong et al., 2021) in our experiments.

Table 1: Test Accuracy (%) of Transformer-MGK compared with the baseline softmax transformer on the LRA benchmark. Our Transform-MGKs outperform softmax transformers while using half the number of heads, having less parameters, and requiring less FLOPs (see Figure 3 for details). Results are averaged over 5 runs.

| Model | ListOps | Text | Retrieval | Average |
|---|---|---|---|---|
| *Softmax 12 heads* | 36.64 | 65.62 | 82.18 | 61.48 |
| *Softmax 8 heads* | 37.03 | **65.71** | 81.74 | 61.49 |
| Transformer-sMGK 4 heads | **37.25** | 65.51 | **82.79** | **61.85** |
| Transformer-MGK 4 heads | 36.98 | 65.69 | 82.23 | 61.63 |
| *Softmax 4 heads* | 36.89 | 65.26 | 81.54 | 61.23 |
| Transformer-sMGK 2 heads | **37.35** | 65.17 | **82.20** | **61.57** |
| Transformer-MGK 2 heads | 36.88 | **65.37** | 81.83 | 61.36 |
| *Softmax 2 heads* | 36.76 | 64.90 | 79.1 | 60.25 |
| Transformer-sMGK 1 head | **37.31** | 65.04 | **81.23** | **61.19** |
| Transformer-MGK 1 head | 37.13 | **65.40** | 80.63 | 61.05 |
| Softmax 1 head | 36.81 | 64.48 | 77.9 | 59.73 |

Table 2: Test Accuracy (%) of Transformer-MLK compared with the linear transformer on the LRA. Our Transform-MLKs achieve comparable/better accuracy than the baselines while using half the number of heads, having less parameters, and requiring less FLOPs (see Figure 3 for details). Results are averaged over 5 runs.

| Model | ListOps | Text | Retrieval | Average |
|---|---|---|---|---|
| *Linear 12 heads* | 20.26 | 65.87 | 81.97 | 56.03 |
| *Linear 8 heads* | 19.17 | **65.85** | 81.18 | 55.40 |
| Transformer-sMLK 4 heads | **20.11** | 65.74 | **81.53** | **55.79** |
| Transformer-MLK 4 heads | 20.06 | 65.7 | 81.34 | 55.7 |
| *Linear 4 heads* | 19.37 | **65.81** | 81.65 | 55.61 |
| Transformer-sMLK 2 heads | 19.88 | 65.61 | **81.66** | **55.71** |
| Transformer-MLK 2 heads | **20.12** | 65.72 | 80.80 | 55.54 |
| *Linear 2 heads* | 18.35 | **65.94** | 80.94 | **55.07** |
| Transformer-sMLK 1 head | **18.87** | 65.57 | 80.37 | 54.93 |
| Transformer-MLK 1 head | 18.34 | 65.70 | **81.09** | 55.04 |
| *Linear 1 head* | 18.60 | 65.70 | 80.6 | 54.96 |

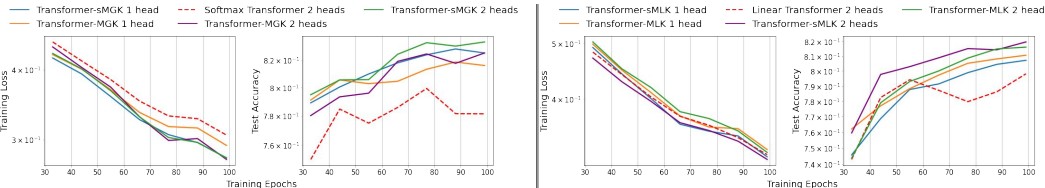

Figure 1: Training loss and test accuracy of Transformer-MGK/MLK vs. softmax/linear transformer on the retrieval task, which has the longest average sequence-length and attention span among the LRA tasks (Tay et al., 2021). The impressive performance of Transformer-MGK/MLK on this challenging task validates the capability of our models to capture long-range dependencies via learning a diversity of attention patterns.

**Results** We summarize our results in Table 1. Transformer-MGKs with half the number of heads consistently achieve better test accuracy than the baseline softmax attention across tasks. Since fewer heads are needed, transformer-MGKs use less parameters and need less FLOPs to compute than the baselines. We provide a detailed efficiency analysis for Transformer-MGKs in Figure 3. More interestingly, these efficiency advantages of Transformer-MGK over the baseline become more significant as the number of heads in the baseline model grows. When using the same number of heads as the baseline models, Transformer-MGKs further improve over those baselines. Among the models, Transformer-sMGK performs the best across LRA tasks.

We also compare the performance of Transformer-MLK with the baseline linear transformers in Table 2. Like Transformer-MGK, Transformer-MLK yields comparable or better results than the baseline using only half the number of heads with less parameters and FLOPs. When using the same number of heads, Transformer-MLK helps improve the linear transformer further.

We provide results of the 12-head baselines in Table 1 and 2 for reference. It is interesting to notice from Table 1 and 2 that even our 2-head Transformer-MGK/MLK models achieve better or equivalent results to the 12-head and 8-head baselines. A comparison between the 12-head baselines with our 6-head Transformer-MGK/MLK models on the retrieval task is provided in Table 8 in Appendix A.9.

In Figure 1, we compare the training loss and test accuracy curves of our 1-head and 2-head Transformer-MGK/MLK with the 2-head softmax and 2-head linear transformers on the document

Table 3: Perplexity (PPL) on WikiText-103 of Transformer-MGK and MLK compared to the baselines. Both Transformer-MGK and MLK achieve comparable or better PPL than the baselines while using only half the number of heads. When using the same number of heads, our models significantly improve the baselines.

| Method | Valid PPL | Test PPL |
|---|---|---|
| *Softmax 8 heads (small)* | 33.15 | 34.29 |
| Transformer-MGK 4 heads (small) | 33.28 | 34.21 |
| Transformer-sMGK 8 heads (small) | 32.92 | 33.99 |
| Transformer-MGK 8 heads (small) | 32.74 | **33.93** |
| *Softmax 4 heads (small)* | 34.80 | 35.85 |
| *Linear 8 heads (small)* | 38.07 | 39.08 |
| Transformer-MLK 4 heads (small) | 38.49 | 39.46 |
| Transformer-MLK 8 heads (small) | 37.78 | **38.99** |
| *Linear 4 heads (small)* | 39.32 | 40.17 |
| *Softmax 8 heads (medium)* | 27.90 (Schlag et al., 2021) | 29.60 (Schlag et al., 2021) |
| Transformer-MGK 4 heads (medium) | 27.58 | **28.86** |

retrieval task. This retrieval task has the longest average sequence-length and attention span among the LRA tasks (Tay et al., 2021). On this task, as shown in Figure 1, our Transformer-MGKs/MLKs are always better than the baseline models throughout the training. This observation corroborates our models's capability of capturing long-range dependencies in very long input sequences.

## 3.2 LANGUAGE MODELING ON WIKITEXT-103

Next we confirm the advantage of our models on a large-scale application. We consider the word-level language modeling task on WikiText-103 (Merity et al., 2017) for our experiments in this section.

**Models and baselines** We compare 4 and 8-head Transformer-MGKs/MLKs with 8-head softmax (Vaswani et al., 2017) and linear transformers (Katharopoulos et al., 2020). Each model consists of 16 layers. Our experiments follow the setting for small/medium models from (Schlag et al., 2021).

**Results** As shown in Table 3, our Transformer-MGKs outperform the baseline softmax transformers. Even when using half the number of attention heads (i.e., 4 vs. 8 heads as in the baselines), the Transformer-MGK still achieves better test perplexities (PPL) than the baseline. Adding more heads into Transformer-MGKs improves their performance. Similarly, Transformer-MLKs attain comparable test/validation PPL to the baseline linear transformers when using half the number of attention heads. When using the same number of attention heads as in the baseline, Transformer-MLKs consistently achieve better performance. Note that reducing the number of heads from 8 to 4 in the baseline models significantly decreases their performance with more than 1.5 reduction in test/validation PPL for the softmax transformer and more than 1.0 reduction in test/validation PPL for the linear transformer. Our proposed Transformer-MGK and Transformer-MLK helps close this gap.

To further examine the scalability of our models, we apply the MGK on a stronger baseline, which is the 8-head medium softmax transformer in (Schlag et al., 2021). This model has 90M parameters, 16 layers, 8 attention heads per layer, and hidden size of 256. The size of our baseline model is close to BERT$_{Base}$ (Devlin et al., 2019), which has 110M parameters, 12 layers, 12 attention heads per layer, and hidden size of 768. Applying our MGK on top of this baseline and using only 4 heads instead of 8, we significantly improve the test PPL from 29.60 to 28.86 while reducing the model size and computational cost, demonstrating the advantages of our scaled models.

## 3.3 NEURAL MACHINE TRANSLATION ON IWSLT'14 GERMAN TO ENGLISH

We further examine the advantages of our methods on the IWSLT'14 German-English machine translation task (Cettolo et al., 2014). Table 9 in Appendix A.10 shows that the 2-head Transformer-MGK and sMGK models achieve the BLEU score of 34.34 and 34.69, respectively, which are comparable to and better than the BLUE score of 34.42 of the 4-head softmax transformer baseline.

## 3.4 EMPIRICAL ANALYSIS

We conduct empirical analysis based on the Transformer-MGK trained for the document retrieval tasks. Results for Transformer-MLKs and the WikiText-103 task are provided in the Appendix.

**Transformer-MGK helps avoid learning redundant heads** We visually compare attention matrices learned by Transformer-MGKs and the baseline softmax transformer on the document retrieval task in Figure 2. In particular, we randomly select an attention matrix at each head in each layer and visualize that attention matrix for each model in comparison. Figure 2(Left) shows that the queries

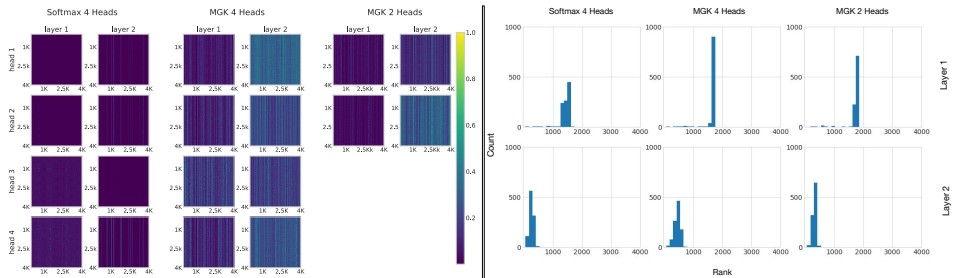

Figure 2: (Left) Visualization of attention matrices in the baseline 4-head softmax transformer (left), 4-head Transformer-MGK (middle), and 2-head Transformer-MGK (right) trained on the document retrieval task. Attention matrices from our Transformer-MGKs have more diverse pattern than those from the baseline softmax transformer, reducing the risk of learning redundant heads. (Right) Rank distribution of attention matrices shows that attention matrices in Transformer-MGK have higher rank than those in the softmax transformer and thus can capture more diverse attention patterns.

in Transformer-MGKs can attend to a variety of keys and equivalently to other tokens at different positions in the input sequence. This diversity in attention pattern helps reduce the chance that the model learns similar and redundant attention matrices at different heads significantly.

Another metric to measure the representation capacity of an attention matrix is its rank. Attention matrices with high ranks can capture more diverse attention patterns compared to those with low ranks (Nguyen et al., 2021). We study the rank of the attention matrix from the Transformer-MGK and the softmax transformer trained for the retrieval task. In particular, we randomly select 1000 different attention matrices at each layer from each model. Then, we perform singular value decomposition (SVD) to compute the rank of each matrix and threshold the singular values smaller than $10^{-6}$. Figure 2(Right) presents the distribution of the rank of attention matrices at each layer of the Transformer-MGK and the softmax transformer. We observe that attention matrices in Transformer-MGK has higher rank than those in the softmax transformer. Thus, our attention with MGK is capable of capturing more diverse and complex attention patterns than the baseline softmax attention.

**Transformer-MGK/MLK reduces model complexity and computational cost** Figure 3 compares the computational cost, measured in FLOPS, and model complexity, measured in the number of parameters, between our Transformer-MGK/MLK that has half the number of heads and the full-head softmax/linear transformer. The more heads being used, the more advantage Transformer-MGK/MLK has over the softmax/linear transformer. For much larger transformer models, this saving is significant.

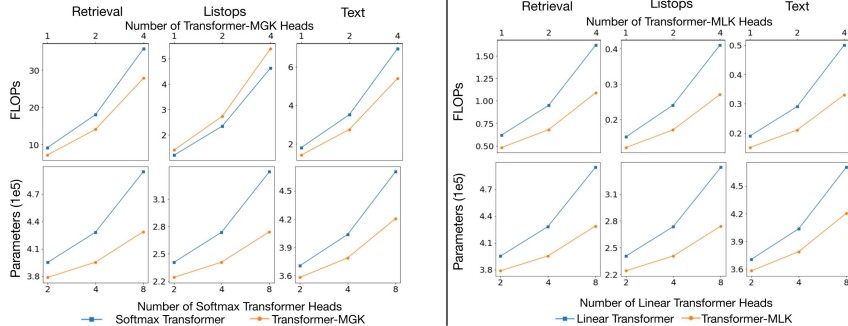

Figure 3: Computational cost (FLOPs) and the number of parameters of Transformer-MGK vs the baseline softmax transformer (Left) and Transformer-MLK vs. the baseline linear transformer (Right). The efficiency advantage of Transformer-MGK/MLK over the baselines in both metrics grows with the number of head.

**Comparing different inference and learning techniques** Table 4 compares the performance of Transformer-MGKs using different design options mentioned in Section 2.3 on the LRA benchmark. In particular, we consider the following three design options: A) Soft-E step, parameters $\pi_{jr}$ and $\boldsymbol{k}_{jr}$ are learnable via SGD, and variance $\sigma_{jr}^2$ are constants, B) Soft-E step, parameter $\pi_{jr}$ is updated according to the M-step update, $\boldsymbol{k}_{jr}$ are learnable via SGD, and variance $\sigma_{jr}^2$ are constants, and C) Hard-E step, $\pi_{jr}$ and $\boldsymbol{k}_{jr}$ are learnable via SGD, and variance $\sigma_{jr}^2$ are constants. Note that Transformer-MGKs with setting A are the default models we use in all experiments above. In Table 4, Transformer-MGK + Hard-E is the Transformer-MGK with setting C, Transformer-MGK + Soft-E is the Transformer-MGK with setting B, and Transformer-MGK only is the Transformer-MGK with

Table 4: Performance of Transformer-MGK using different inference/learning techniques on LRA benchmark.

| Model | ListOps | Text | Retrieval | Average |
|---|---|---|---|---|
| Transformer-sMGK + Hard-E 1 head | 37.25 | 64.7 | 81.29 | 61.08 |
| Transformer-sMGK + Soft-E 1 head | 37.05 | 64.68 | **81.44** | 61.05 |
| Transformer-sMGK 1 head | **37.31** | **65.04** | 81.23 | **61.19** |
| Transformer-MGK + Hard-E 1 head | 19.40 | **65.40** | 80.72 | 55.17 |
| Transformer-MGK + Soft-E 1 head | 33.85 | 65.25 | **80.73** | 59.94 |
| Transformer-MGK 1 head | **37.13** | **65.40** | 80.63 | **61.05** |

setting A. It is worth noting that Transformer-sMGK + Hard-E obtains comparable results to the models with the best performance in each task even though it is the most efficient model in our study.

## 4 RELATED WORK

**Efficient Transformers** Efficient transformers can be classified into several categories, as summarized in (Roy et al., 2021). Among these categories are models which design the attention matrix to have sparse structure (Parmar et al., 2018; Liu et al., 2018; Qiu et al., 2019; Child et al., 2019; Beltagy et al., 2020). Another category includes models that combine two or more different access patterns to improve the coverage (Child et al., 2019; Ho et al., 2019). Access patterns can also be made learnable in a data-driven fashion (Kitaev et al., 2020; Roy et al., 2021; Tay et al., 2020). Other efficient transformers take advantage of a side memory module to access multiple tokens at once (Lee et al., 2019; Sukhbaatar et al., 2019; Asai & Choi, 2020; Beltagy et al., 2020). Finally, low-rank and kernelization approximation are utilized to enhance the memory and computational efficiency of computing self-attention (Tsai et al., 2019; Wang et al., 2020; Katharopoulos et al., 2020; Choromanski et al., 2021; Shen et al., 2021; Nguyen et al., 2021; Peng et al., 2021). In addition to the aforementioned efficient transformers, multi-query attention that shares keys and values between different attention heads (Shazeer, 2019) has also been studied to reduce the memory-bandwidth cost and increase the speed for incremental transformer inference (see Appendix A.11). Last but not least, methods such as using auxiliary losses (Al-Rfou et al., 2019) and adaptive input embedding (Baevski & Auli, 2019) have been explored to speed up the convergence of training transformers. Our MGK/MLK can be easily incorporated into the methods above to further improve their accuracy and efficiency.

**Redundancy in Transformers** Latest works suggest that most of the neurons and heads in the pre-trained transformer are redundant and can be removed when optimzing towards a downstream task (Dalvi et al., 2020; Michel et al., 2019; Durrani et al., 2020). Other works also study the contextualized embeddings in pretrained networks under this redundancy due to overparameterization and show that the representations learned within these models are highly anisotropic (Mu & Viswanath, 2018; Ethayarajh, 2019). An emerging body of work is proposed to distill and prune the model, including (Sanh et al., 2019; Sun et al., 2019; Voita et al., 2019b; Sajjad et al., 2020). Our MGK/MLK approch can be combined with these distilling and pruning methods to improve their performance.

**Mixture Models for Transformers** Several works have used mixture models to study and enhance transformers. Switch transformers (Fedus et al., 2021) employ the routing algorithm in Mixture of Experts (MoE) to reduce the communication and computational costs in transformers. (Nguyen et al., 2018; Patel et al., 2016) derive a probablistic framework based on GMMs for deep neural networks that can be extended to study transformers and attention-based architectures. Other works that use mixture models with transformers include (Cho et al., 2020; Guo et al., 2019; Jiang et al., 2020).

## 5 CONCLUDING REMARKS

In this paper, we proposed Transformer-MGK, a class of transformers that use Gaussian mixture model to represent the key vectors in self-attention. Transformer-MGK reduces the redundancy among heads in transformer. Furthermore, attention heads in the Transformer-MGK have better representation capability than those in the baseline, allowing the Transformer-MGK to achieve comparable or better performance than the baseline softmax transformer while using only half of the number of heads. Comparing to the baseline, the Transformer-MGK uses fewer parameters and requires less FLOPs to compute. We extend the Transformer-MGK into the Transformer-MLK to use linear attentions for better efficiency. We empirically validate the advantage of the Transformer-MGK/MLK over the baseline softmax and linear transformers on various benchmarks including tasks in the LRA benchmark, WikiText-103 language modeling, and IWSLT'14 machine translation. In our work, we make the means and the variance of the cluster learnable variables and constants, respectively. It is interesting to explore how to leverage the M-step update in the EM algorithm to update those parameters. Furthermore, we leave the application of Transformer-MGK/MLK for improving the vision transformer (Dosovitskiy et al., 2020; Touvron et al., 2020) as future work.

**Reproducibility Statement:** Source codes for our experiments are provided in the supplementary materials of the paper. The details of our experimental settings and computational infrastructure are given in Section 3 and the Appendix. All datasets that we used in the paper are published, and they are easy to find in the Internet.

**Ethics Statement:** Given the nature of the work, we do not foresee any negative societal and ethical impacts of our work.

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

# Supplement to "Transformer with a Mixture of Gaussian Keys"

In this supplementary material, we provide experimental details and additional experiments of Transformer-MGK and Transformer-MLK.

## A ADDITIONAL EXPERIMENTS

### A.1 EXPERIMENT DETAILS

In this section, we provide model and training details for experiments in Section 3. All our experiments are conducted on a server with 4 NVIDIA A100 GPUs.

#### A.1.1 LONG RANGE ARENA BENCHMARK

**Datasets and metrics** We consider the following tasks in the LRA benchmark: Listops (Nangia & Bowman, 2018), byte-level IMDb reviews text classification (Maas et al., 2011), and byte-level document retrieval (Radev et al., 2013). These tasks involve long sequences of length $2K$, $4K$, and $4K$, respectively. We follow the setup/evaluation protocol in (Tay et al., 2021) and report the test accuracy for individual task and the average result across all tasks.

**Models and baselines** We use the softmax transformer (Vaswani et al., 2017) and linear transformer (Katharopoulos et al., 2020) as our baselines. All models have 2 layers, 64 embedding dimension, and 128 hidden dimension. The number of heads in each layer are set to 1, 2, 4, and 8. For Transformer-MGK/MLKs and their shifted versions, we share $\pi_{jr}$ for all position $j$ and learn it for each head. The initial value for each $\pi_{jr}$ is set to 0.5. For Transformer-sMGK, we learn $\mathbf{b_r}$ and initialize its elements from a standard normal distribution. Each $\sigma_{jr}$ is a constant with value $\sqrt{D}$ where $D$ is the dimension of each head, which is the same as in the baselines models

Details about the Long Range Arena (LRA) benchmarks can be found in the original paper (Tay et al., 2021). Our implementation is based on the public code by (Xiong et al., 2021), and we follow their training procedures. The training setting and additional baseline model details are provided in the configuration file used in (Xiong et al., 2021) and available at https://github.com/mlpen/Nystromformer/blob/main/LRA/code.

#### A.1.2 LANGUAGE MODELING ON WIKITEXT-103

**Datasets and metrics** WikiText-103 consists of articles from Wikipedia and is a dataset with long contextual dependencies. The training set is made up of about $28K$ articles containing $103M$ running words; this corresponds to text blocks of about 3600 words. The validation and test sets are composed of $218K$ and $246K$ running words, respectively. Each of them contains 60 articles and about $268K$ words. Our experiment follows the standard setting (Merity et al., 2017; Schlag et al., 2021) and splits the training data into $L$-word independent long segments. For evaluation, we use a batch size of 1, and go through the text sequence with a sliding window of size $L$. We consider only the last position for computing perplexity (PPL) except in the first segment, where all positions are evaluated as in (Al-Rfou et al., 2019; Schlag et al., 2021).

**Models and baselines** Our language modeling implementation is based on the public code https://github.com/IDSIA/lmtool-fwp by (Schlag et al., 2021). We use their small and medium model configurations for models in our experiments. In particular, for small models, we set the key, value, and query dimension to 128, and the training and evaluation context length to 256. For medium models, we set the key, value, and query dimension to 256, and the training and evaluation context length to 384. In both configurations, we set the number of heads to 8, the feed-forward layer dimension to 2048, and the number of layers to 16. For Transformer-MGK/MLK, we use our 4 and 8-head versions to compare with the 8-head baselines. $\pi_{ir}$, $\mathbf{b_r}$ and $\sigma_{jr}$ for this task follow our setings for LRA experiments. Other than those, our language modeling share the same configurations as the baselines.

We train our models for language modeling on 2 A100, 40GB each with a batch size of 96, and each model is trained for 120 epochs. We apply 10% dropout (Hanson, 1990; Srivastava et al., 2014) and use the Adam optimizer (Kingma & Ba, 2014) with an initial learning rate of 0.00025 and 2000 steps for learning rate warm-up.

#### A.1.3 NEURAL MACHINE TRANSLATION ON IWSLT'14 GERMAN TO ENGLISH

**Datasets and metrics** The IWSLT14 German-English dataset (Cettolo et al., 2014) contains $153K$ training, $7K$ validation, and $7K$ test TED-talks scripts German-English translated sentences. We

follow the same preprocessing steps as in (Ott et al., 2019). We use the BiLingual Evaluation Understudy (BLEU) score as our evaluation metric for this task. All trained models are evaluated using the evaluation protocol in (Ott et al., 2019).

**Models and baselines** The baseline model we use in this machine translation task is an encoder-decoder transformer with six encoder/decoder layers, four attention heads per layer. The embedding dimension and the hidden size are 512 and 1024, respectively, for both encoder and decoder. These architecture configurations are the same for Transformer-MGK/sMGK except for the number of attention heads per layer, which is reduced by half. We share $\pi_{jr}$ across all position $j$ and learn it for each head. The initial value for each $\pi_{jr}$ is set to 0.5. For Transformer-sMGK, we learn $\boldsymbol{b}_r$ and initialize its elements from a standard normal distribution. Each $\sigma_{jr}$ is a constant with the value $\sqrt{D}$ where $D$ is the dimension of each head, which is the same as in the baselines models. Our experiments follow the settings from (Ott et al., 2019), and our implementation is based on the public code https://github.com/pytorch/fairseq/tree/main/examples/translation.

### A.2 MORE EMPIRICAL ANALYSIS OF TRANSFORMER-MGKS/MLKS TRAINED FOR LANGUAGE MODELING

In Table 3 in the main text, we show the improvements in valid and test perplexity of our Transformer-MGKs/MLKs compared with the baseline softmax and linear transformers. In particular, Transformer-MGK/MLKs with the same number of heads as the baselines, e.g. 8 heads, significantly improve the baselines during training while Transformer-MGK/MLKs with 4 head achieve comparable or better performance than the 8-head baselines. In this section, we provide more empirical analysis to shed light on those results. Figure 4 shows the validation perplexity curve of our models versus the softmax and linear transformers.

Figure 5 visualizes the attention matrices from a randomly selected sample for the trained softmax transformer with 8 heads and Transformer-MGKs with 8 heads and 4 heads. These visualizations show that Transformer-MGKs attend to more diverse positions in all heads and layers than the softmax attention baseline. We also compare the rank distributions of these attention matrices computed from 1000 samples at each layer in the model. Figure 6 presents the rank histograms of the 8-head softmax attention and 4-head and 8-head MGK attentions for the $1^{st}$ and $5^{th}$ layer. It is clear that attention matrices from the Transformer-MGKs have higher ranks than those in the softmax transformer, which implies that Transformer-MGK can attend to more diverse regions than the softmax transformer without the need of using more attention heads.

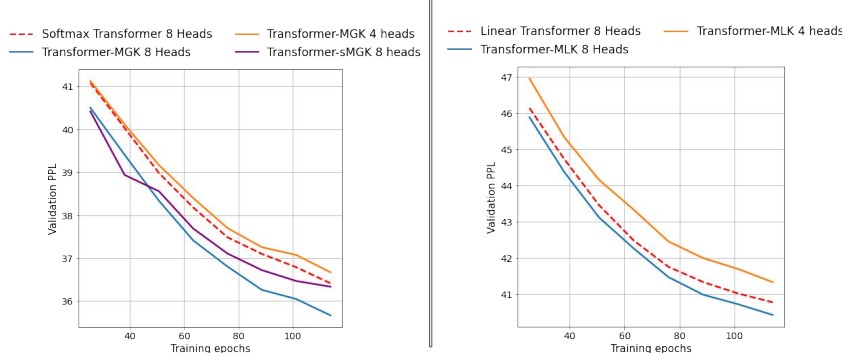

Figure 4: Validation perplexity of Transformer-MGK vs. the softmax transformer (Left) and Transformer-MLK vs. the linear transformer (Right) for language modeling on WikiText-103.
.

### A.3 ADDITIONAL TRAINING RESULTS FOR LRA

In this section, we provide additional experimental results on the LRA benchmark. Figure 7 compares the computational cost measured by FLOPs and model complexity in term of the number of parameters of different inference and learning methods for Transformer-MGK. The computational costs of Transformer-MGK/sMGK and Transformer-MGK Hard-E/Soft-E are on par with each other, while Transformer-sMGK uses fewer parameters than the other without trade-off in performance 4 for all tasks. The naming is as explained in Section 3.4 in the main text. In addition, Figure 8 visualizes the attention matrices in the 4-head linear transformer baseline, 4-head Transformer-MLK, and 2-head Transformer-MLK trained on the document retrieval task.

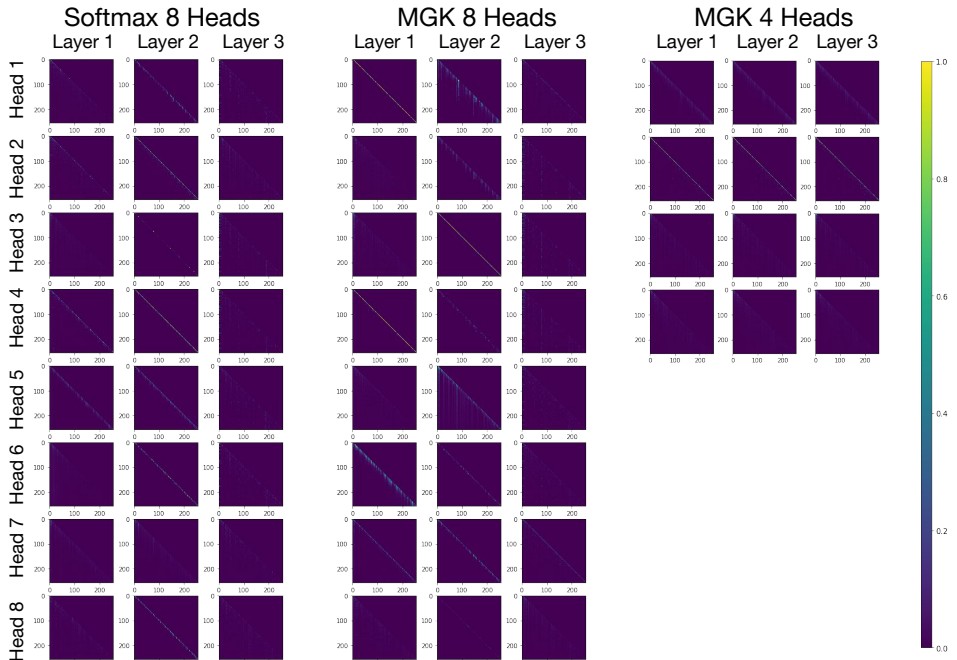

Figure 5: Visualization of attention matrices from 8-head softmax transformer (Left), 8-head Transformer-MGK (Middle), and 4-head Transformer-MGK (Right) trained on WikiText-103 language modeling. Here, we plot the attention matrices for all heads and layers in the models. The sequence length is 256 and the size of each matrix is 256×256.

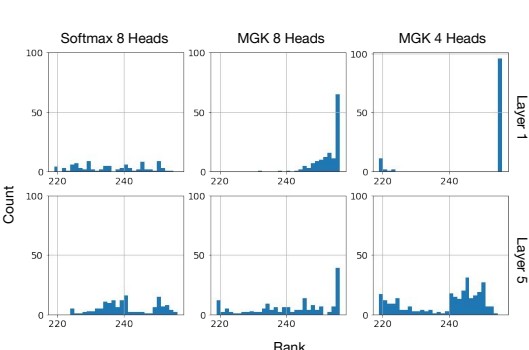

Figure 6: Rank distributions of attention matrices from 8-head softmax transformer (Left), 8-head Transformer-MGK (Middle), and 4-head Transformer-MGK (Right) trained on WikiText-103 language modeling. The rank histograms are computed from 1000 attention matrices at each layer. The attention matrices from Transformer-MGKs have higher ranks than those from softmax transformers. This implies that Transformer-MGKs have more diverse attention patterns, which allows us to reduce the number of heads in Transformer-MGKs.

## A.4 Ablation Study on the Impact of the Mixture of Keys, the Gaussian Distance, and the Key Shifting

In this section, we conduct an ablation study of the Transformer-MGK on the LRA retrieval task to investigate where the performance improvement is from. In particular, we would like to understand the impact of the following factors on the performance of Transformer-MGK: 1) the mixture of keys, 2) the Gaussian distance, and 3) the key shifting. We summarize our empirical results in Table 5 and discuss the impact of 1, 2 and 3 below.

**Impact of the Mixture of Keys** We apply our mixture of keys (MGK) approach to the softmax transformer using the dot product between queries and keys instead of the Gaussian distance as in our paper. We name this model Softmax MGK. We compare the Softmax MGK that has 1 head (Sofmax MGK 1 head in Table 5) with the baseline softmax transformers that use 1 and 2 heads (Softmax 1 head and Softmax 2 heads in Table 5). Results in Table 5 show that the Softmax MGK 1 head outperforms both the baseline softmax transformers of 1 and 2 heads. Note that our Softmax MGK

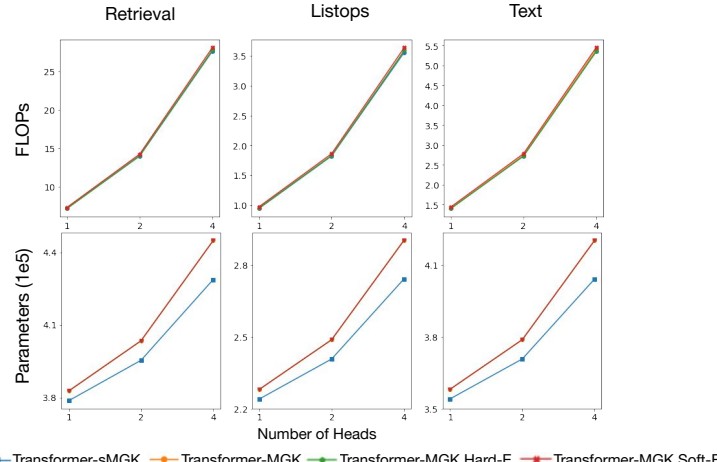

Figure 7: Model complexity (Top) and computational cost (Bottom) of different inference and learning methods for Transformer-MGK trained on the document retrieval task. While computational costs are almost the same, Transformer-sMGK has more advantage in model size, comparing to Transformer-MGK, Transformer-MGK Hard-E, and Soft-E. The naming is as explained in Section 3.4 in the main text.

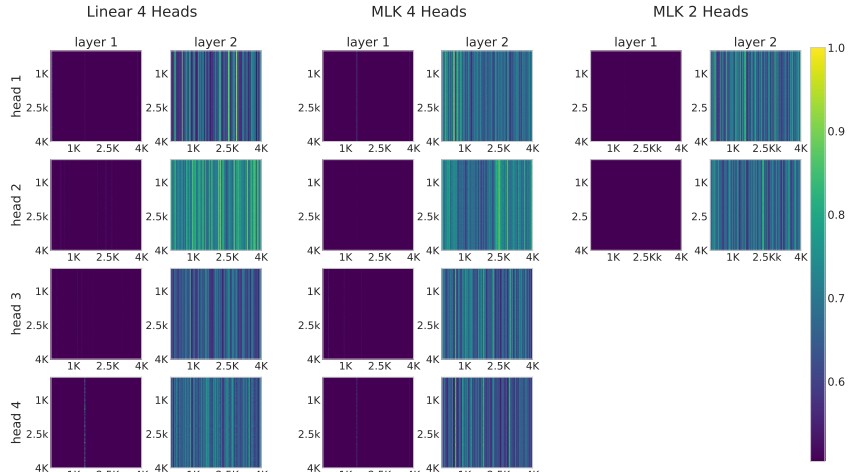

Figure 8: Visualization of attention matrices in the 4-head linear transformer baseline (Left), 4-head Transformer-MLK (Middle), and 2-head Transformer-MLK (Right) trained on the document retrieval task. Here, the sequence length is 4000, and the size of each matrix is 4000×4000.

1 head is more efficient than the baseline of 2 heads in terms of the number of parameters and the number of FLOPs. These results confirm the benefit of using MGK.

**Impact of using the Gaussian distance** Next, we compare the softmax MGK with the Gaussian MGK. Here the Gaussian MGK is the Transformer-MGK proposed and discussed in our paper, which computes the attention scores using the MGK approach and the Gaussian distance between the queries and keys. Results in Table 5 suggest that the Gaussian MGK 1 head improves over the Softmax MGK 1 head (80.63% vs. 79.23%). This result justifies the advantage of using Gaussian distance over dot product to compute the attention scores.

**Impact of key shifting** Finally, we apply key shifting to both Softmax MGK and Gaussian MGK (Softmax sMGK and Gaussian sMGK in Table 5). From Table 5, we observe that the Softmax sMGK 1 head and Gaussian sMGK 1 head outperform the Softmax MGK 1 head and Gaussian MGK 1 head, respectively. These results, again, corroborate the benefit of using key shifting.

We also include the result for the Gaussian 1 head model in Table 5. This Gaussian 1 head model is similar to the Softmax 1 head model but uses the Gaussian distance to compute the attention scores. Comparing the results of the Gaussian 1 head model, the Softmax 1 head model, and the Gaussian

Table 5: Ablation study on the impact of the mixture of keys, the Gaussian distance, and the key shifting on the LRA retrieval task. We denote the softmax Transformer by Softmax. All Softmax models (i.e., Softmax 2 heads, Softmax 1 head, Softmax MGK 1 head, and Softmax sMGK 1 head) use dot product to compute the attention scores. All Gaussian models (i.e., Gaussian MGK 1 head, Gaussian sMGK 1 head, and Gaussian 1 head) use Gaussian distance to compute the attention scores. We denote MGK with key shifting by sMGK. Here MGK is used to denote our approach of using a mixture of keys at each timestep.

| Method | Accuracy (%) |
| --- | --- |
| *Softmax 2 heads* | 79.10 |
| Softmax sMGK 1 head | **79.81** |
| Softmax MGK 1 head | 79.23 |
| *Softmax 1 head* | 77.90 |
| Gaussian sMGK 1 head | **81.23** |
| Gaussian MGK 1 head | 80.63 |
| *Gaussian 1 head* | 80.38 |

Table 6: Comparing the GPU memory footprint and computational time overhead (seconds/iteration) between our 4-head Transformer-MGKs/MLKs and the 8-head softmax/linear transformer baselines trained on the LRA retrieval task at test time. Our Transformer-MGKs/MLKs save much more memory and have significantly smaller wall-clock time compared to the baselines. Here we use a batch size of 32 for each iteration.

| Method | Memory (Gb) | Time overhead (seconds/iteration) |
| --- | --- | --- |
| *Softmax 8 heads* | 58.00 | 0.357 |
| Transformer-MGK 4 heads | 53.58 | 0.278 |
| Transformer-sMGK 4 heads | **45.50** | **0.227** |
| *Linear 8 heads* | 3.38 | 0.055 |
| Transformer-MLK 4 heads | 2.90 | 0.043 |
| Transformer-sMLK 4 heads | **2.83** | **0.042** |

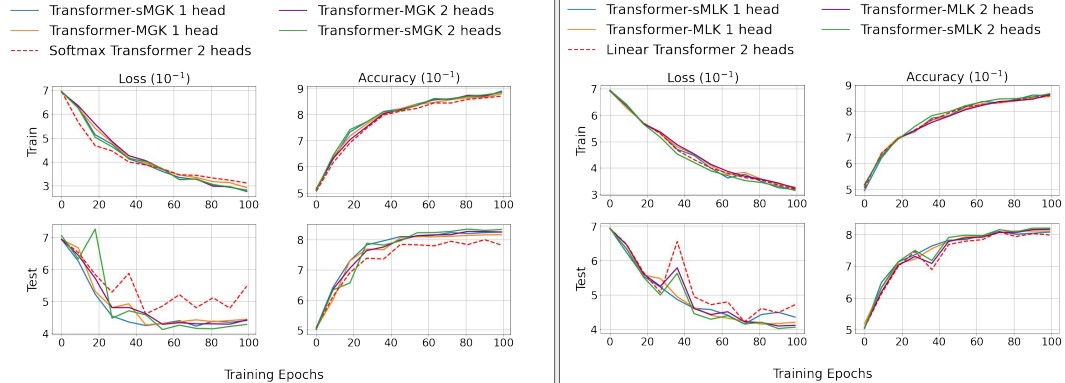

Figure 9: Training and test loss/accuracy of Transformer-MGK vs. softmax transformer (Left) and Transformer-MLK vs. linear Transformer (Right) on the retrieval task, which has the longest average sequence-length and attention span among the LRA tasks (Tay et al., 2021). In training, we apply early stopping to avoid overfitting. That explains why the training loss and accuracy curves stop early. The test loss/accuracy curves already converge. The impressive performance of Transformer-MGK/MLK on this challenging task validates the capability of our models to capture long-range dependencies via learning a diversity of attention patterns.

MGK 1 head model reported in Table 5 further confirms the advantage of using MGK and Gaussian distance.

## A.5 TIME OVERHEAD AND MEMORY FOOTPRINT ANALYSIS

Table 6 compares the GPU memory footprint and computational time overhead (seconds/iteration) of our 4-head Transformer-MGKs/MLKs with those of the 8-head softmax/linear transformer baselines at test time. All models are trained on the LRA retrieval task, and we use a batch size of 32 for each iteration. Our MGK/MLK models save memory and reduce wall-clock time significantly compared to the baselines. Using key shifting in our models, i.e. Transformer-sMGKs/sMLKs, helps improve the efficiency further.

## A.6 LEARNING CURVES SHOWING CONVERGENCE

In this section, we replot Figure 1 and 4 to show the convergence of our trainings. In Figure 1, the results do not seem to converge since we plot the loss and the accuracy in log-scale. In Figure 4, the

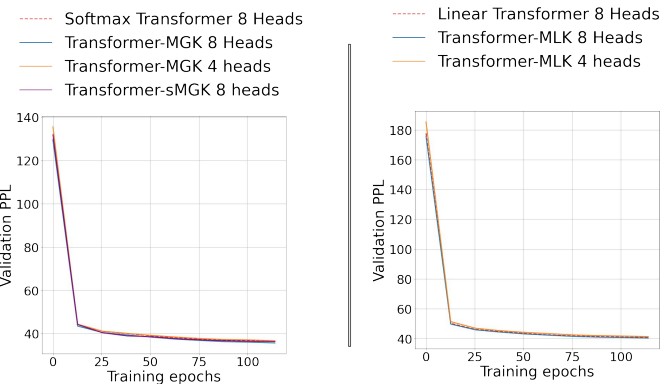

Figure 10: Validation perplexity of the Transformer-MGK vs. the softmax transformer (Left) and the Transformer-MLK vs. the linear transformer (Right) for language modeling on WikiText-103. Training converges on this task after 500000 iterations, equivalent to 115 epochs .

.

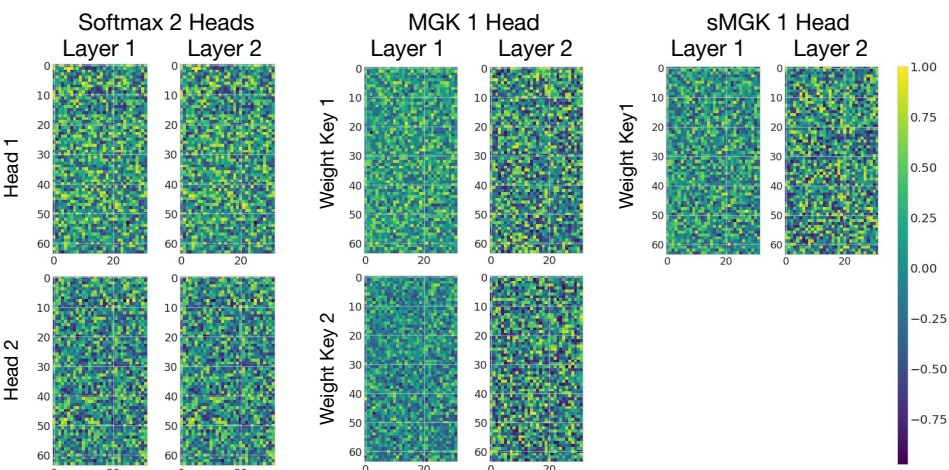

Figure 11: Weight matrices $\mathbf{W}_K$ for computing the keys, for all heads and layers, in the 2-head softmax transformer baseline (Left), the 1-head Transformer-MGK with 2 keys (Middle), and the 1-head Transformer-sMGK with 2 keys (Right) trained on the LRA retrieval task. Here, the dimension of each head $D = 32$ and that of input $x_i$ is $D_x = 64$. Hence, each weight matrix has the shape of $(64, 32)$.

Table 7: The learned mixing coefficient $\pi_{jr}$ of all heads and layers in the 1-head Transformer-MGKs trained on the LRA retrieval task. Here we use the same $\pi_{j1}, \pi_{j2}$ for all time step $j = 1, \ldots, N$.

| Method | Layer 1 | | Layer 2 | |
|---|---|---|---|---|
| | $\pi_{j1}$ | $\pi_{j2}$ | $\pi_{j1}$ | $\pi_{j2}$ |
| Transformer-sMGK 1 heads | 0.488 | 0.512 | 0.500 | 0.500 |
| Transformer-MGK 1 heads | 0.502 | 0.498 | 0.497 | 0.503 |

results do not seem to converge since we zoom into the specific range on the y and x-axes. Figure 9 and 10 are the replotted versions of Figure 1 and 4, respectively. In Figure 9, the training loss/accuracy curves stop early because we use early stopping to avoid overfitting. The test loss/accuracy curves in this figure already converge.

### A.7 WEIGHT MATRICES OF THE KEYS, KEYS AND MIXING COEFFICIENT

In this section, we analyze the learned $\pi_{jr}$, $\boldsymbol{k}_{jr}$, and $\mathbf{W}_{K_r}$, $j = 1, \ldots, N$ and $r = 1, \ldots, M$ in the Transformer-MGK trained on the LRA retrieval task. In all of our experiments, we set $M = 2$. In Figure 11 and 12, we visualize the weight matrices $\mathbf{W}_K$ that computes the keys and the keys $\mathbf{K}$, respectively, for all heads and layers in the 2-head softmax transformer baseline, the 1-head Transformer-MGK with 2 keys, and the 1-head Transformer-sMGK with 2 keys trained on the LRA retrieval task. Note that for the keys, we only plot the first 100 tokens. Also, Table 7 summarizes the

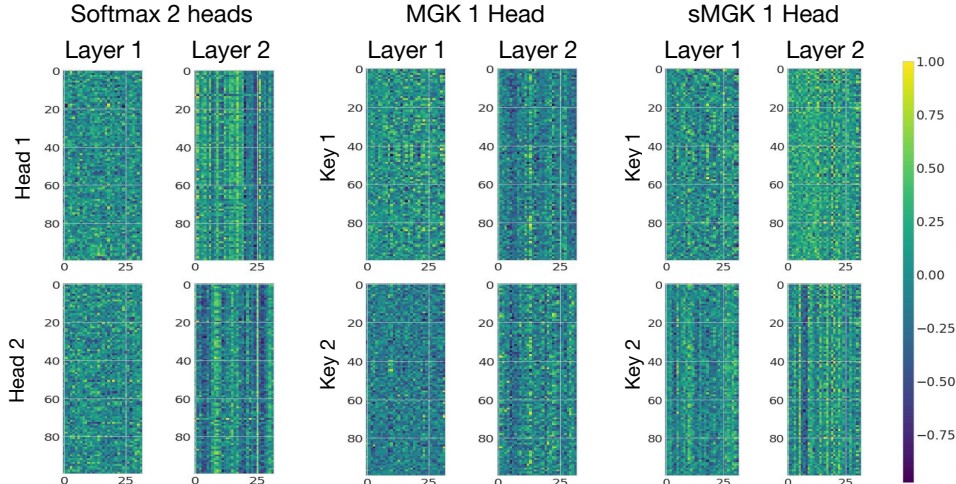

Figure 12: Key embeddings **K** for all heads and layers of the 2-head softmax transformer baseline (Left), the 1-head Transformer-MGK with 2 keys (Middle), and the 1-head Transformer-sMGK with 2 keys (Right) trained on the LRA retrieval task. Here the dimension $D$ of each head is 32, and we plot the key embeddings of the first 100 tokens in a randomly chosen sequence. Hence, each key matrix has the shape of $(100, 32)$.

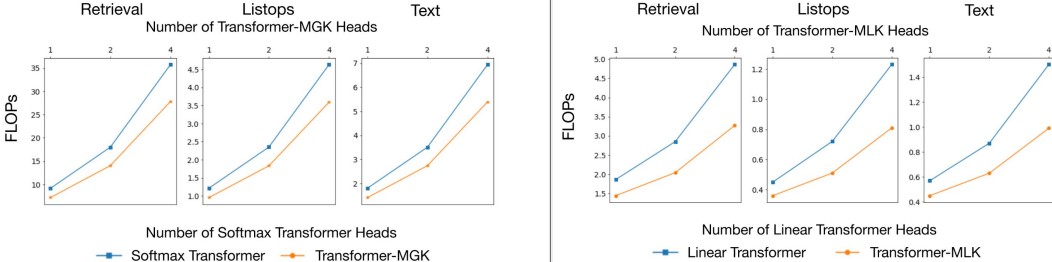

Figure 13: Computational cost (FLOPs) for each training iteration of Transformer-MGK vs. the baseline softmax transformer (Left) and Transformer-MLK vs. the baseline linear transformer (Right) on the LRA retrieval task. The efficiency advantage of Transformer-MGK/MLK over the baselines grows with the number of heads. Here, the batch size of each training iteration is 32.

learned mixing coefficient $\pi_{jr}$ of all heads and layers in the 1-head Transformer-MGK trained on the same retrieval task. Here we use the same $\pi_{j1}, \pi_{j2}$ for all time step $j = 1, \ldots, N$.

### A.8 ADDITIONAL COMPUTATIONAL COMPLEXITY (FLOPS) ANALYSIS AT TRAINING TIME

Figure 13 demonstrates the computational cost (FLOPs) for each training iteration of the Transformer-MGK vs. the baseline softmax transformer (Left) and the Transformer-MLK vs. the baseline linear transformer (Right) on the LRA retrieval task. The efficiency advantage of Transformer-MGK/MLK over the baselines grows with the number of heads.

Figure 14 shows the computational cost per training iteration (measured in FLOPs) of different inference and learning methods for Transformer-MGK trained on the document retrieval task. Transformer-sMGK, Transformer-MGK, Transformer-MGK Hard-E, and Soft-E have similar computational costs.

### A.9 SCALING TO 12-HEAD BASELINE MODELS FOR THE RETRIEVAL TASK.

To further study the scalability of our model, in this section, we investigate the performance of our 6-head Transformer-MGKs/MLKs in comparison with the 12-head baseline softmax/linear transformers on the retrieval task. Table 8 indicates that our 6-head Transform-MGKs/MLKs significantly outperform 12-head softmax/linear transformers, respectively. Moreover, comparing these results to those in Table 1 and Table 2, although the 12-head softmax/linear transformers improve over the 8-head ones, their accuracies are still worse than or only equivalent to those of our 4-head and even 2-head Transformer-MGK/MLK models.

### A.10 NEURAL MACHINE TRANSLATION ON IWSLT'14 GERMAN TO ENGLISH

Table 9 shows that the 2-head Transformer-MGK/sMGK models have comparable or better BLEU scores than the 4-head softmax transformer baseline.

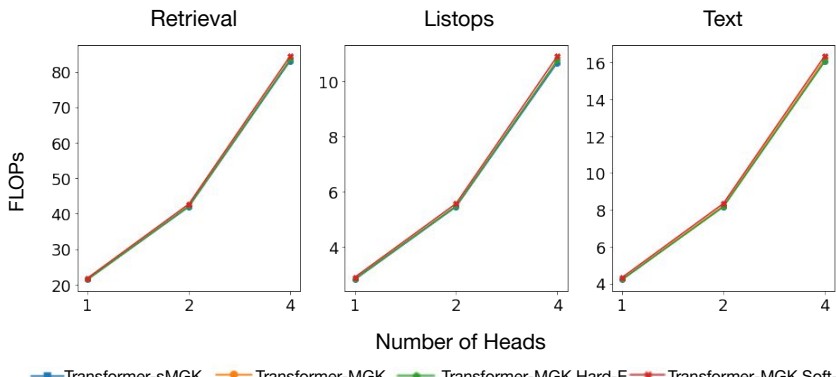

Figure 14: Computational cost (measured in FLOPs) per training iteration of different inference and learning methods for Transformer-MGK trained on the LRA retrieval task. Computational costs are almost the same for Transformer-sMGK, Transformer-MGK, Transformer-MGK Hard-E, and Soft-E. The naming is as explained in Section 3.4 in the main text.

Table 8: Test Accuracy (%) of 6-head Transformer-MGKs/MLKs compared with the baseline 12-head softmax and linear transformers on the retrieval task. Our 6-head Transformer-MGKs/MLKs significantly outperform softmax and linear transformers, respectively, while being more efficient in terms of computational cost, model size, and memory usage.

| Method | Accuracy (%) |
|---|---|
| *Softmax 12 heads* | 82.18 |
| Transformer sMGK 6 head | **83.31** |
| Transformer MGK 6 head | 83.05 |
| *Linear sMGK 12 head* | 81.97 |
| Transformer sMLK 6 head | **82.80** |
| Transformer MLK 6 head | 82.11 |

Table 9: Machine translation BLEU scores of 2-head Transformer-MGKs on the IWSLT14 De-En dataset is better than or equivalent to that of the 4-head baseline.

| Method | BLEU score |
|---|---|
| *Softmax 4 heads* | 34.42 |
| Transformer sMGK 2 head | **34.69** |
| Transformer MGK 2 head | 34.34 |

## A.11 COMPARISON TO MULTI-QUERY ATTENTION

In this section, we compare our MGK approach to the multi-query attention (Shazeer, 2019). The multi-query attention shares the same set of keys and values at different heads to reduce the memory-bandwidth cost during incremental inference, which allows faster inference since the size of the reloaded "keys" and "values" tensors are significantly reduced. On another hand, our Transformer-MGK models the key at each head as a Gaussian mixture model, which leads to the use of multiple keys at each head and allows us to decrease the number of attention heads. This helps reduce the computations and parameters needed to calculate additional queries and values. If using key shifting (see option (B) in paragraph Design Options for Keys in Section 2.3), the MGK approach also helps reduce the computations and parameters needed to calculate additional keys. The advantages of Transformer-MGK hold in both training and inference, including incremental inference. We have provided a detailed analysis on the computational complexity and the number of parameters of the Transformer-MGK in comparison with the corresponding softmax transformer in Appendix B. Combining the multi-query attention and our MGK approach is interesting since each method has its own advantage and they are complementary to each other. In particular, we can let the transformer model share the same set of values and mixtures of keys at different heads. This approach can potentially have the advantages of both multi-query attention and MGK.

## B  An Analysis on the Computational Complexity and the Number of Parameters in Transformer-MGK and the Softmax Transformer

In this section, we compare the computational complexity and the number of parameters in transformer-MGK with $M$ keys at each timestep and $H/M$ heads to the baseline softmax transformer that has 1 key at each timestep and $H$ heads. Here we choose $M$ such that $H$ is a multiple of $M$ and use keys design option (A) that make the key $k_{jr}$ a linear projection of the input $x_j$, i.e. $k_{jr} = x_j \mathbf{W}_{K_r}^\top$, where $x_j \in \mathbb{R}^{1 \times D_x}$, $\mathbf{W}_{K_r} \in \mathbb{R}^{D \times D_x}$ and $r = 1, 2, \ldots, M$ (see Design Options for Keys in Section 2.3 for more details). To simplify notation and without loss of generalization, we let $M = 2$ as in our experiments and assume that $D_v = D$, i.e., the values have the same feature dimension as the queries and the keys. To simplify the computation, we also do not take the softmax operator into account since this operator yields similar costs when applied in Transformer-MGK and softmax transformer.

### B.1  Computational Complexity

**(i) Softmax $H$-head attention:** The number of computations in a softmax $H$-head attention is $N^2 H(4D - 1) + NHD(6D_x + 2HD - 5)$.

*Explanation:* To calculate the query matrix $\mathbf{Q}$, the key matrix $\mathbf{K}$, and the value matrix $\mathbf{V}$ in Step 1 in Section 1.1 at each head, we need $3NDD_x$ multiplications and $3ND(D_x - 1)$ additions. In total, these need $3ND(2D_x - 1)$ computations. Next, to compute the product $\mathbf{Q}\mathbf{K}^\top$ in Eqn. (1), we need $N^2 D$ multiplications and $N^2(D - 1)$ additions. Similarly, the product $\mathbf{A}\mathbf{V}$ requires $N^2 D$ multiplications and $N(N - 1)D$ additions. In total, computing the output sequence $\mathbf{H}$ in Eqn. (1) at each head requires $3ND(2D_x - 1) + N^2 D + N^2(D - 1) + N^2 D + N(N - 1)D = N^2(4D - 1) + ND(6D_x - 4)$ computations. The total computation for all $H$ heads is then

$$H(N^2(4D - 1) + ND(6D_x - 4)) + NHD(2HD - 1)$$
$$= N^2 H(4D - 1) + NHD(6D_x + 2HD - 5),$$

where the extra $NHD(2HD - 1)$ is from the linear projection by $\mathbf{W}^O$.

**(ii) Mixture of 2 Gaussian keys attention with $H/2$-head:** The number of computations in a mixture of 2 Gaussian keys attention with $H/2$-head is $N^2 H(3D - 0.5) + NHD(4D_x + HD - 4)$.

*Explanation:* Similar to the above derivation, in a Mixture of M Gaussian keys attention, to compute the output sequence $\mathbf{H}$ we need $N^2((2M + 2)D - 1) + ND((M + 2)(2D_x - 1) - 1)$ computations. Note that, to compute the Gaussian distances between the queries $q_i$ and the keys $k_j$ as in the mixture of M Gaussian keys attention and to compute their dot product as in the softmax attention, we need the similar number of computations. Therefore, the total computation for all $H/M$ heads is then

$$(H/M)(N^2((2M + 2)D - 1) + ND((M + 2)(2D_x - 1) - 1)) + NHD(2(H/M)D - 1)$$
$$= N^2 H \left( \frac{2(M + 1)D - 1}{M} \right) + NHD \left( \frac{2(M + 2)}{M} D_x + \frac{2}{M} HD - \frac{3M + 2}{M} \right).$$

Let $M = 2$, then the total computation of the mixture of 2 Gaussian keys attention is given by $N^2 H(3D - 0.5) + NHD(4D_x + HD - 4)$.

**Soft-max H-head attention versus mixture of 2 Gaussian keys attention with H/2-head:** Given the results in (i) and (ii), when compared to the baseline softmax $H$-head attention, our mixture of 2 Gaussian keys attention with $H/2$-head saves

$$N^2 H(D - 0.5) + NHD(2D_x + HD - 1)$$

computations. When $N$ is large, this difference is significant. In conclusion, the mixture of 2 Gaussian keys attention with H/2-heads has cheaper computational complexity than that of soft-max H-head attention.

### B.2  The Number of Parameters

**(iii) Softmax $H$-head attention:** The number of parameters in a softmax $H$-head attention is $3HDD_x + (HD)^2$.

*Explanation:* $3HDD_x$ is from the linear projects to calculate the query matrix $\mathbf{Q}$, the key matrix $\mathbf{K}$, and the value matrix $\mathbf{V}$ in Step 1 in Section 1.1. $(HD)^2$ is from the linear project to compute the final output as in Eqn. (3).

**(iv) Mixture of 2 Gaussian Keys attention with $H/2$-head:** The number of parameters in a Mixture of 2 Gaussian Keys attention with $H/2$-head is $2HDD_x + 0.5(HD)^2 + H$.

*Explanation:* The linear projections to calculate $\mathbf{Q}$ and $\mathbf{V}$ contribute $HDD_x/2$ parameters each. The linear projection to calculate $\mathbf{K}$ contributes $HDD_x$. The linear project to compute the final output has dimension $(H/2)D \times HD$, so it contributes $0.5(HD)^2$ parameters. $H$ more parameters is from the prior $\pi_{jr}$. These priors contribute 2 parameters at each head since we make $\{\pi_{j1}, \pi_{j2}\}$ for all $j = 1, \ldots, N$ the same.

**Soft-max H-head attention versus mixture of 2 Gaussian keys attention with H/2-head:** Given the results in (iii) and (iv), when compared to the baseline softmax $H$-head attention, our mixture of 2 Gaussian keys attention with $H/2$-head saves $HDD_x + 0.5(HD)^2 - H$ parameters. When $H$ and $D$ are large, this saving is significant.

## C    Proofs of main results

In this appendix, we provide proofs for the main results in the paper.

### C.1    Proof of Theorem 1

To ease the presentation of the proof, for any probability distribution $G$, we denote

$$p_G(x) := \int f(x - \theta)dG(\theta) = \int \phi(x|\theta, \sigma^2 \mathbf{I})dG(\theta),$$

for all $x \in \mathbb{R}^d$ where $f(x) = \frac{1}{(\sqrt{2\pi}\sigma)^d} \exp\left(-\frac{\|x\|^2}{2\sigma^2}\right)$ for given $\sigma > 0$. It means that $p_G$ is the convolution of $f$ and the probability distribution $G$. Since the space of Gaussian mixtures is dense in the space of continuous probability measures (Bacharoglou, 2010), it indicates that there exists probability distribution $G_1$ such that

$$\sup_{x \in \mathbb{R}^d} |p(x) - p_{G_1}(x)| \leq \frac{\epsilon}{2}. \tag{14}$$

Our next step is to prove that there exists a probability measure $G_2$ with at most $K$ supports where $K \leq (C\log(1/\epsilon))^d$ for some universal constant $C$ such that

$$\sup_{x \in \mathbb{R}^d} |p_{G_1}(x) - p_{G_2}(x)| \leq \frac{\epsilon}{2}. \tag{15}$$

Indeed, from Lemma A.1 in (Ghosal & van der Vaart, 2001), for any $k \geq 1$ there exists a probability distribution $G_2$ with at most $(2k - 2)^d$ supports such that

$$\int \theta^\alpha d(G_1 - G_2)(\theta) = 0, \tag{16}$$

for any $\alpha = (\alpha_1, \alpha_2, \ldots, \alpha_d) \in \mathbb{N}^d$ such that $0 \leq |\alpha| = \sum_{j=1}^d \alpha_j \leq 2k - 2$, Here, $\theta^\alpha = \prod_{j=1}^d \theta_j^{\alpha_j}$.

Now, for any $M \geq 2a\sqrt{d}$, we have $\|x - \theta\| \geq \|x\| - \|\theta\| > M - a\sqrt{d} > M/2$ as long as $\|x\| > M$ and $\theta \in [-a, a]^d$. It indicates that

$$\sup_{\|x\|>M} |p_{G_1}(x) - p_{G_2}(x)| = \sup_{\|x\|>M} \left| \int f(x - \theta)d(G_1 - G_2)(\theta) \right|$$

$$\leq \sup_{\|x\|>M} \int \frac{1}{(\sqrt{2\pi}\sigma)^d} \exp\left(-\frac{\|x - \theta\|^2}{2\sigma^2}\right) d(G_1 + G_2)(\theta)$$

$$\leq \frac{2}{(\sqrt{2\pi}\sigma)^d} \exp\left(-\frac{M^2}{8\sigma^2}\right). \tag{17}$$

On the other hand, for any $k \geq 1$ we also have that

$$
\begin{aligned}
\sup_{\|x\| \leq M} |p_{G_1}(x) - p_{G_2}(x)| &= \sup_{\|x\| \leq M} \left| \int f(x - \theta) d(G_1 - G_2)(\theta) \right| \\
&\leq \sup_{\|x\| \leq M} \left| \int \left( f(x - \theta) - \sum_{j=0}^{k-1} \frac{(-1)^j \|x - \theta\|^{2j}}{(\sqrt{2\pi})^d \sigma^{d+2j} j!} \right) d(G_1 - G_2)(\theta) \right| \\
&\quad + \sup_{\|x\| \leq M} \left| \int \sum_{j=0}^{k-1} \frac{(-1)^j \|x - \theta\|^{2j}}{(\sqrt{2\pi})^d \sigma^{d+2j} j!} d(G_1 - G_2)(\theta) \right| \\
&= \sup_{\|x\| \leq M} \left| \int \left( f(x - \theta) - \sum_{j=0}^{k-1} \frac{(-1)^j \|x - \theta\|^{2j}}{(\sqrt{2\pi})^d \sigma^{d+2j} j!} \right) d(G_1 - G_2)(\theta) \right|,
\end{aligned}
$$
(18)

where the final equality is stems from

$$
\int \sum_{j=0}^{k-1} \frac{(-1)^j \|x - \theta\|^{2j}}{(\sqrt{2\pi})^d \sigma^{d+2j} j!} d(G_1 - G_2)(\theta) = 0,
$$

which is due to Eqn. (16).

To further bound the right-hand-side (RHS) of Eqn. (18), we use the following inequality:

$$
\left| \exp(y) - \sum_{j=0}^{k-1} (y)^j / j! \right| \leq |y|^k / k!
$$

for any $y \in \mathbb{R}$. Since $k! \geq (k/e)^k$ for any $k \geq 1$, the above bound can be rewritten as

$$
\left| \exp(y) - \sum_{j=0}^{k-1} (y)^j / j! \right| \leq \frac{|ye|^k}{k^k}.
$$
(19)

Further simplification of Eqn. (18) leads to

$$
\begin{aligned}
\sup_{\|x\| \leq M} |p_{G_1}(x) - p_{G_2}(x)| &\leq \sup_{\|x\| \leq M} \int \left| f(x - \theta) - \sum_{j=0}^{k-1} \frac{(-1)^j \|x - \theta\|^{2j}}{(\sqrt{2\pi})^d \sigma^{d+2j} j!} \right| d(G_1 + G_2)(\theta) \\
&\leq 2 \sup_{\|x\| \leq M, \theta \in [-a,a]^d} \left| f(x - \theta) - \sum_{j=0}^{k-1} \frac{(-1)^j \|x - \theta\|^{2j}}{(\sqrt{2\pi})^d \sigma^{d+2j} j!} \right| \\
&= \sup_{\|x\| \leq M, \theta \in [-a,a]^d} \frac{2}{(\sqrt{2\pi}\sigma)^d} \left| \exp\left( -\frac{\|x - \theta\|^2}{2\sigma^2} \right) - \sum_{j=0}^{k-1} \frac{(-1)^j \|x - \theta\|^{2j}}{\sigma^{2j} j!} \right| \\
&\leq \sup_{\|x\| \leq M, \theta \in [-a,a]^d} \frac{e^k \|x - \theta\|^{2k}}{\sigma^{2k} (2k)^k},
\end{aligned}
$$

where the final inequality is based on an application of inequality (19) with $y = -\|x - \theta\|^2 / (2\sigma^2)$. For $\|x\| \leq M$ and $\theta \in [-a, a]^d$, we have $\|x - \theta\| \leq \|x\| + \|\theta\| \leq M + a\sqrt{d}$. Therefore, we further have

$$
\sup_{\|x\| \leq M} |p_{G_1}(x) - p_{G_2}(x)| \leq \sup_{\|x\| \leq M, \theta \in [-a,a]^d} \frac{e^k \|x - \theta\|^{2k}}{\sigma^{2k} (2k)^k} \leq \frac{e^k (M + a\sqrt{d})^{2k}}{\sigma^{2k} (2k)^k}.
$$

When $M \geq 2a\sqrt{d}$, we have $M + a\sqrt{d} \leq \frac{3M}{2}$ and the above bound leads to

$$
\sup_{\|x\| \leq M} |p_{G_1}(x) - p_{G_2}(x)| \leq \frac{(9e)^k M^{2k}}{(8\sigma^2 k)^k}.
$$
(20)

By choosing $M^2 = 8\sigma^2 \log(1/\epsilon')$ for some $\epsilon' > 0$, the bounds in Eqns. (17) and (20) become

$$\sup_{\|x\| \leq M} |p_{G_1}(x) - p_{G_2}(x)| \leq \frac{2}{(\sqrt{2\pi}\sigma)^d}\epsilon',$$

$$\sup_{\|x\| > M} |p_{G_1}(x) - p_{G_2}(x)| \leq \frac{(9e)^k(\log(1/\epsilon'))^k}{k^k}. \tag{21}$$

As long as we choose $k = 9e^2 \log(1/\epsilon')$ and $\epsilon' \leq 1$, we have

$$\sup_{\|x\| > M} |p_{G_1}(x) - p_{G_2}(x)| \leq e^{-k} = e^{-9e^2 \log(1/\epsilon')} = (\epsilon')^{9e^2} \leq \epsilon'. \tag{22}$$

By choosing $\epsilon' = \frac{\epsilon}{2\max\{\frac{2}{(\sqrt{2\pi}\sigma)^d}, 1\}}$, the results from Eqns. (21) and (22) indicate that

$$\sup_{\|x\| \leq M} |p_{G_1}(x) - p_{G_2}(x)| \leq \frac{\epsilon}{2}, \quad \text{and} \quad \sup_{\|x\| > M} |p_{G_1}(x) - p_{G_2}(x)| \leq \frac{\epsilon}{2}.$$

Therefore, if we choose $M = 8\sigma^2 \log\left(\frac{2\max\{\frac{2}{(\sqrt{2\pi}\sigma)^d}, 1\}}{\epsilon}\right)$ and $k = 9e^2 \log\left(\frac{2\max\{\frac{2}{(\sqrt{2\pi}\sigma)^d}, 1\}}{\epsilon}\right)$, we have

$$\sup_{x \in \mathbb{R}^d} |p_{G_1}(x) - p_{G_2}(x)| \leq \frac{\epsilon}{2}.$$

It indicates that we obtain the conclusion of claim (15) by choosing $K = (2k - 2)^d \leq \left(18e^2 \log\left(\frac{2\max\{\frac{2}{(\sqrt{2\pi}\sigma)^d}, 1\}}{\epsilon}\right)\right)^d$. Combining the results from Eqns. (14) and (15), we have

$$\sup_{x \in \mathbb{R}^d} |p(x) - p_{G_2}(x)| \leq \sup_{x \in \mathbb{R}^d} |p(x) - p_{G_1}(x)| + \sup_{x \in \mathbb{R}^d} |p_{G_1}(x) - p_{G_2}(x)| \leq \epsilon.$$

As a consequence, we obtain the conclusion of the theorem.

