# OpenReview forum: "Transformer with a Mixture of Gaussian Keys"
_ICLR.cc/2022/Conference — ICLR 2022 Submitted_

### Official Review · Reviewer_fA4U · 2021-10-24

**Correctness:** 3
**Technical Novelty And Significance:** 3
**Empirical Novelty And Significance:** 3
**Recommendation:** 6
**Confidence:** 4

**Main Review:**

Strengths:
The authors use the probabilistic framework to establish the connection between self-attention and gaussian distribution. Even though there are some assumptions: (i) the query and the key are normalized; (ii) all the standard derivations are identity; (iii) the prior distribution is uniform, I think these assumptions are easy to follow. Based on the connection, a natural extension is the mixture Gaussian model and they apply the EM algorithm to learn the posterior distribution. The experiment results are comprehensive.

Weaknesses:
1. In Table 1 and Table 2, the authors discover the influence of the number of heads. However, [1] claims that one head is indeed sufficient, which is not consistent with your results. I wonder whether the reason is the experiment settings.
2. In Table 1, Transformer-sMGK is better than Transformer-MGK in some cases but not in all cases. Can you provide some explanation about the reason? Also the situation in Table 2.
3. In Table 1 and Table 2, the authors start from 8 heads. Can you provide the results from the vanilla Transformer (12 heads) for better illustration?

[1] Michel, Paul, Omer Levy, and Graham Neubig. "Are Sixteen Heads Really Better than One?." Advances in Neural Information Processing Systems 32 (2019): 14014-14024.


**Summary Of The Paper:**

This paper proposes Transformer with a Mixture of Gaussian Keys (Transformer-MGK), which replaces the multi-head self-attention module with a Gaussian mixture model. Specifically, the authors build the connection between self-attention and probabilistic framework. Based on their findings, they extend the self-attention to a Gaussian mixture attention. Compared with vanilla Transformer, Transformer-MGK has fewer parameters and accelerates training and inference with comparable performance. Various experiment results illustrate the advantage of Transformer-MGK.

**Summary Of The Review:**

This paper builds the connection between self-attention and Gaussian distribution with some assumptions, which is novel for Transfomer model. The extension from original self-attention to mixture Gaussian model is natural and convincing to me. The experiment results are sufficient to illustrate the improvement of their models.

---

> ### Author Response · Authors · 2021-11-15
> **Response to Reviewer fA4U**
>
> Thank you for your thoughtful review and valuable feedback. Below we address your concerns.
>
> -----
>
> **Q1. In Table 1 and Table 2, the authors discover the influence of the number of heads. However, [1] claims that one head is indeed sufficient, which is not consistent with your results. I wonder whether the reason is the experiment settings.
> [1] Michel, Paul, Omer Levy, and Graham Neubig. "Are Sixteen Heads Really Better than One?." Advances in Neural Information Processing Systems 32 (2019): 14014-14024.**
>
> **Reply:** [1] does head pruning after training. While one head can be sufficient at test time after pruning, more heads are needed during the training so that a good local minimum can be found easier. This is consistent with findings in neural network pruning that the sparse network uncovered by pruning are harder to train from the start, reaching lower accuracy than the original networks [2, 3, 4].
>
> ​​[2] Hao Li, Asim Kadav, Igor Durdanovic, Hanan Samet, and Hans Peter Graf. Pruning filters for efficient convnets. ICLR, 2016.
>
> [3] Song Han, Jeff Pool, John Tran, and William Dally. Learning both weights and connections for efficient neural network. NeurIPS, 2015.
>
> [4] Jonathan Frankle and Michael Carbin. The lottery ticket hypothesis: Finding sparse, trainable neural networks. ICLR, 2019.
>
> -----
>
> **Q2. In Table 1, Transformer-sMGK is better than Transformer-MGK in some cases but not in all cases. Can you provide some explanation about the reason? Also the situation in Table 2.**
>
> **Reply:** Transformer-MGK and Transformer-sMGK, whose keys are shifted versions of each other, are two design options for keys in our MGK models. Transformer-sMGK uses fewer parameters and requires less computation than Transformer-MGK. Which design option yields better results is task-dependent. For the task that requires models with more expressive power, the Transformer-MGK with more parameters can be a better choice. For the task that requires models with more structure, the Transformer-sMGK can be a better choice. In practice, it is hard to identify exactly what type of models a task needs. Thus, for a new task, it is good to try both Transformer-MGK and Transformer-sMGK and select the one that yields better results.
>
> -----
>
> **Q3. In Table 1 and Table 2, the authors start from 8 heads. Can you provide the results from the vanilla Transformer (12 heads) for better illustration?**
>
> **Reply:** Thanks for your suggestion. We will provide the results from the vanilla transformer with 12 heads in our revised manuscript soon.
>
> -----
>
>
> We hope we have cleared your concerns about our work. We have also revised our manuscript according to your comments, and we would appreciate it if we can get your further feedback at your earliest convenience.

---

> > ### Author Response · Authors · 2021-11-21
> > **Response to Reviewer fA4U  - New Empirical Results to Address Q3**
> >
> > **Q3. In Table 1 and Table 2, the authors start from 8 heads. Can you provide the results from the vanilla Transformer (12 heads) for better illustration?**
> >
> > **Reply:** As the reviewer suggests, we have added results for 12-head  softmax/linear transformers (vanilla) trained on the LRA benchmark into Tables 1 and 2, respectively, in our revised manuscript. Our 4-head Transformer-MGK and Transformer-MLK still achieve better or equivalent results to the baseline 12-head softmax/linear transformers across the ListOps, text classification, and retrieval tasks. In particular, the 4-head Transformer-MGK and Transformer-MLK achieve the average accuracy of 61.85\% and 55.79\%, respectively, on these three tasks compared to 61.48\% and 56.03\% of 12-head softmax and linear transformers, respectively. It is interesting to notice from Table 1 and 2 that even our 2-head Transformer-MGK/MLK models achieve better or equivalent results to the 12-head and 8-head baselines. Note that compared to the 12-head/8-head baselines, our 4-head/2-head models are much more efficient in terms of the model size, memory usage, and computational cost. We also provide additional results for 6-head Transformer-MGK and Transformer-MLK for the retrieval task in comparison with the 12-head softmax and linear transformers in Table 8 in Appendix A.9 of the revised manuscript. Our 6-head Transformer-MGK/MLK models significantly outperform the 12-head baseline in this task. We also note that for the LRA tasks, in most cases (except for the linear transformer trained on the ListOps task), increasing the number of heads from 8 to 12 in the baseline does not help much in improving the accuracy of the trained models. We hypothesize that this might be due to the redundancy between heads and overfitting.

---

> > > ### Comment · Reviewer_fA4U · 2021-11-22
> > > **Thank you for the response**
> > >
> > > I have read all comments and the author's response.
> > > My main concerns have been addressed and I keep my original score.

---

> > > > ### Author Response · Authors · 2021-11-23
> > > > **Thanks for your endorsement!**
> > > >
> > > > Thanks for your response and we appreciate your endorsement.

---

### Official Review · Reviewer_qyjK · 2021-11-02

**Correctness:** 3
**Technical Novelty And Significance:** 3
**Empirical Novelty And Significance:** 2
**Recommendation:** 6
**Confidence:** 3

**Main Review:**

 Strong points

- The idea of introducing the mixture of Gaussian keys to increase the expression capacity of key embeddings and to allow the query to attend to more diverse positions is interesting.
- Overall, the paper is well written. Especially, in section 2, the formula explanations have a neat flow so that it is easy to follow.
- The proposed model shows comparable results to general softmax-based transformers with fewer parameters and FLOPs.
- In the experiment analysis, the visualization of the attention matrix and rank distribution supports that the paper's hypothesis that Transformer-GMK would learn diverse attention patterns.

Weak points

- The main concern about this paper is the experimental improvements are marginal. Moreover, although the method shows comparable to original transformers with fewer parameters and FLOPs, I suggest the authors to discuss whether there is any difficulty in the learning process using EM-based algorithms for practical usability; for example, how many learning epochs are required, whether it is sensitive to initial values, and an increase of computation required to calculate the E-step during the training process.
- Beyond the attention matrix analysis, in order to demonstrate the keys are well modeled as Gaussian mixtures, the learned $\pi_{jr}$,  $k_{jr}$, or $W^k_r$ which is the weight matrix projecting to $k_{jr}$ seems to be required to be analyzed.
- In the experiment section, the number of Gaussian mixtures is not noted. Also, comparing with baseline methods (Softmax and Linear heads) and proposing methods with half heads is, for me, unfair without considering the number of the mixtures.
-  I'm not convinced with the experimental results are reported at their convergence since the learning curves in Figures 1 and 4 end before converging.

Questions

- How does the proposed model result with the increasing number of mixtures?
- How many epochs are needed to converge via the EM algorithm?
- Also FLOPs of training time including the E-steps should be compared.
- Can this proposed approach be applied to deeper and large Transformers such as BERT and GPTs and show its effectiveness?

Comments

- I would like to recommend comparing the model complexity and computational cost with notations and formulas rather than empirical figures.
- Typo in Figure 3. The label of Softmax Transformer is wrong as Liner Transformer in the left figure.
- Typo in Table 4. The lower rows group: Transformer-MGK → Transformer-MLK

**Summary Of The Paper:**

This paper proposes a transformer with a mixture of Gaussian keys (Transformer-MGK) in order to avoid training redundant attention heads in the transformer modules. The authors formulate the attention mechanism as a posterior distribution $p(t_j = 1 | q_i)$ (i.e., given the query $q_i$, how likely $q_i$ matches the key $k_j$), and extend this as the mixture of Gaussian keys where each key $k_j$ is modeled as a mixture of $M$ Gaussians with several assumptions. The paper introduces inference methods using the EM algorithm: Soft E-step and Hard E-step. The authors also propose a liner variant, Transformer-MLK, which can be more computationally efficient to long sequences. Empirical studies are performed to show the effectiveness of Transformer-MGK over the softmax Transformers. Results are shown on the long-range arena benchmark and language model on WikiText-103 corpus. Also, comparisons of the model complexity and computational cost and ablation study depending on different inference and learning techniques are also provided.

**Summary Of The Review:**

Overall, I vote for "marginally below the acceptance threshold". I think the extension of attention keys to the mixture of Gaussian and exploring the learning methods with EM algorithms are somewhat technically sound. However, experimentally demonstrating its practical usefulness is required further. Hopefully, the authors can address my concern in the rebuttal period.

---

> ### Author Response · Authors · 2021-11-15
> **Response to Reviewer qyjK (1/3)**
>
> Thank you for your thoughtful review and valuable feedback. Below we address your concerns.
>
> -----
>
> **Q1. The main concern about this paper is the experimental improvements are marginal.**
>
> **Reply:** We believe there is a misunderstanding of the advantage of our Transformer-MGK/MLK over the baselines. Please allow us to clear this misunderstanding by clarifying the key results from our experiments. In our experiments on LRA and Wikitext-103,  we set **$M = 2$** for Transformer-MGK/MLK, i.e. using only 2 keys at each timestep, and show that:
>
>
> (1) **Transformer-MGK/MLK with half the number of heads is comparable or better than the baseline softmax and linear transformers with the full number of heads** while being more efficient in both computational cost and memory footprint.
>
>
> (2) **Using the same number of heads, Transformer-MGK/MLK significantly outperforms the baseline softmax and linear transformers.**
>
>
> On Point (1), Tables 1, 2, and 3 in our paper verify that our Transformer-MGKs/MLKs with 1 head, 2 heads, and 4 heads outperform or at least on par with the baseline softmax and linear transformers with double the number of heads, i.e., 2 heads, 4 heads, and 8 heads, respectively, on LRA and Wikitext-103. It is interesting to notice from Table 1 and 2 that even our 2-head Transformer-MGK/MLK models achieve better or equivalent results to the 12-head and 8-head baselines on the LRA benchmark. Using only half the number of heads or even fewer than that, our Transformer-MGKs/MLKs need much fewer parameters and FLOPs to compute than the baseline models. We believe this advantage of our models is significant.
>
>
> On Point (2), Tables 1, 2, and 3 confirm that our Transformer-MGKs/MLKs with the same number of heads as the baseline models outperform the baselines. In particular, on the task that requires the very long-range attention such as the retrieval task, our Transformer-MGKs/MLKs significantly outperform the baselines.  Note that this retrieval task has the longest average sequence-length and attention span among the LRA tasks (Tay, Yi, et al. Long range arena: A benchmark for efficient transformers. ICLR 2021). Also, on the Wikitext-103 task, our Transformer-MGKs/MLKs with 4 heads significantly outperform the baselines with 4 heads. The advantage gaps between our Transformer-MGKs/MLKs with 8 heads and the baselines with 8 heads on the Wikitext-103 task are smaller because the baselines with 8 heads, given their capacity, already perform very well on this Wikitext-103 task. That says, the improvement in accuracy gained by Transformer-MGKs/MLKs in this case is indeed relevant.

---

> > ### Author Response · Authors · 2021-11-15
> > **Response to Reviewer qyjK (2/3)**
> >
> > **Q2. Moreover, although the method shows comparable to original transformers with fewer parameters and FLOPs, I suggest the authors to discuss whether there is any difficulty in the learning process using EM-based algorithms for practical usability; for example, how many learning epochs are required, whether it is sensitive to initial values, and an increase of computation required to calculate the E-step during the training process.**
> >
> > **Reply:** The E-step inference of our Gaussian mixture model proposed in Eq. 4 matches the computation of self-attention in transformers. Thus, it is as efficient as self-attention in transformers, and no increase of computation is required to calculate the E-step. When using a mixture of Gaussian keys at each time step as in Eq. 7, for example, a mixture of 2 Gaussian keys ($M=2$), we can reduce the number of heads in the model by half compared to the baseline. As a result, the E-step in Transformer-MGKs/MLKs is even more efficient in terms of the number of parameters and FLOPs than self-attention in the baseline transformers.
> >
> >
> > We use the same number of learning epochs as the baseline transformers. Sensitivity to initial values of  $W_{Q}$, $W_{K}$, and $W_{V}$ in our model is the same as how the baseline transformers are sensitive to the initialization of those linear projections. In our experiments, we initialize those matrices from a Gaussian distribution as in the baseline transformers. We initialize the prior parameters $\pi_jr$ as uniform priors, and we observe that this yields the best result. Using other initialization for these prior parameters reduces the accuracy of the trained model.
> >
> > -----
> >
> > **Q3. Beyond the attention matrix analysis, in order to demonstrate the keys are well modeled as Gaussian mixtures, the learned $\pi_{jr}$, $k_{jr}$, or $W_{jr}$ which is the weight matrix projecting to $k_{jr}$ seems to be required to be analyzed.**
> >
> >
> > **Reply:** We have included an analysis of the learned  $\pi_{jr}$, $k_{jr}$, and $W_{K_{r}}$, which is the weight matrix projecting to $k_{jr}$, in Appendix A.7. The results are summarized in Figure 11, Figure 12, and Table 7. Note that $W_{K_{r}}$, $r=1,2$, is the same for all $j=1,2,\dots,N$ in our models (see Design Options for Keys in Section 2.3 in the revised manuscript).
> >
> > -----
> >
> > **Q4. In the experiment section, the number of Gaussian mixtures is not noted. Also, comparing with baseline methods (Softmax and Linear heads) and proposing methods with half heads is, for me, unfair without considering the number of the mixtures.**
> >
> >
> > **Reply:** In all experiments, we set $M = 2$, where $M$ is the number of Gaussians, i.e. keys, at each timestep. Then we compare the baseline softmax and linear transformers with our Transformer-MGKs/MLKs ($M=2$) using half the number of heads and the full number of heads. We’ve made this clear in the second paragraph of section 3 in the revised manuscript.
> >
> > -----
> >
> > **Q5. I'm not convinced with the experimental results are reported at their convergence since the learning curves in Figures 1 and 4 end before converging.**
> >
> >
> > **Reply:** In Figure 1, the results do not seem to converge since we plot the loss and the accuracy in log-scale. In Figure 4, the results do not seem to converge since we zoom into the specific range on the y and x-axes. We have included a version of Figures 1 and 4 without log-scale and zoom, i.e. Figures 9 and 10, in Appendix A.6 and showed that the results reported in those figures converge.

---

> > > ### Author Response · Authors · 2021-11-15
> > > **Response to Reviewer qyjK (3/3)**
> > >
> > > **Q6. How does the proposed model result with the increasing number of mixtures? How many epochs are needed to converge via the EM algorithm? Also FLOPs of training time including the E-steps should be compared.**
> > >
> > > **Reply:** Using the same number of heads, increasing the number of clusters, i.e. keys, in MGK will increase the performance of the model. Also, using more keys allow us to reduce the number of heads more, but there will be a tradeoff since using much fewer heads might lead to difficulty in training the model.
> > >
> > > As mentioned above, we use the same number of learning epochs as the baseline transformers to train all versions of Transformer-MGKs/MLKs (see the Reply to Q2 for more details).
> > >
> > > We have shown FLOPs per training iteration in Figure 13 in Appendix A.8 of our revised manuscript. Transformer-MGK/MLK still significantly outperforms the baseline softmax and linear transformer in this metric. Furthermore, Figure 14 shows FLOPs per training iteration of Transformer-MGKs that use different inference and learning methods. These models have similar computational costs in terms of FLOPs per training iteration.
> > >
> > > -----
> > >
> > > **Q7. Can this proposed approach be applied to deeper and large Transformers such as BERT and GPTs and show its effectiveness?**
> > >
> > > **Reply:** The baseline softmax transformer model we used in our WikiText-103 language modeling experiments has similar structure as BERT. Both our model and BERT are transformers. Compared to the base version of BERT, i.e. BERTBase [1], which has 110M parameters, 12 layers, 12 attention heads per layer, and hidden size of 768, our model has 40M parameters, 16 layers, 8 attention heads per layer, and hidden size of 128. Note that the baseline transformer we used is deeper than BERTBase. Furthermore, comparing to the TinyBERT4 model that can achieve 96.8% performance of BERTBase via knowledge distillation [2], our model is much larger and deeper. In particular, TinyBERT4 only has 4 layer and 14.5M parameters. As a result, we believe that the transformer model we used in our WikiText-103 language modeling experiments is a large enough baseline to show that the advantages of our Transformer-MGK/MLK can be scaled up to deeper and larger models.
> > >
> > > We have also launched additional experiments on WikiText-103 with a larger baseline model. This model has 90M parameters, 16 layers, 8 attention heads per layer, and hidden size of 256. We will include new results with this baseline model in our revised manuscript soon.
> > >
> > > ​​[1]. Jacob Devlin, Ming-Wei Chang, Kenton Lee, and Kristina Toutanova. BERT: Pre-training of Deep Bidirectional Transformers for Language Modeling. NAACL, 2019.
> > >
> > > [2].  Xiaoqi Jiao, Yichun Yin, Lifeng Shang, Xin Jiang, Xiao Chen, Linlin Li, Fang Wang, and Qun Liu. TinyBERT: Distilling BERT for Natural Language Understanding. EMNLP, 2020.
> > >
> > >
> > > -----
> > >
> > > **Q8. I would like to recommend comparing the model complexity and computational cost with notations and formulas rather than empirical figures.**
> > >
> > > **Reply:** We have included a comparison of model complexity and computational cost between the Transformer-MGK and the baseline softmax transformer in Appendix B of our revised manuscript. The Transformer-MGK in our study has $M$ keys at each time step and $H/M$ heads while the baseline softmax transformer has 1 key at each time step and H heads. Compared to the baseline softmax transformer, our Transformer-MGK saves $N^{2}H(D - 0.5) + NHD(2D_{x} + HD - 1)$ computations and $HDD_{x} + 0.5 (HD)^{2} - H$ parameters for each attention, where N is the sequence length, D is the dimension of query/key/value vectors, and D_{x} is the dimension of the input vectors. When $N$, $H$, and $D$ are large, this saving is significant. Note that in our analysis, we assume the value vectors have the same dimension as the query and key vectors to simplify the notation and calculation.
> > >
> > > -----
> > >
> > > **Q9. Typo in Figure 3. The label of Softmax Transformer is wrong as Liner Transformer in the left figure. Typo in Table 4. The lower rows group: Transformer-MGK → Transformer-MLK**
> > >
> > > **Reply:** Thank you so much for pointing these out. We have addressed the typo in Figure 3 in our revision. The lower rows group in Table 4 is for Transformer-MGK, and the upper rows group in Table 4 is for Transformer-sMGK, whose keys are shifted versions of each other, as we mention in the second paragraph of section 3 in our paper.
> > >
> > > -----
> > >
> > > We hope we have cleared your concerns about our work. We have also revised our manuscript according to your comments, and we would appreciate it if we can get your further feedback at your earliest convenience.

---

> > > > ### Author Response · Authors · 2021-11-21
> > > > **Response to Reviewer qyjK - New Empirical Results to Further Address Q1 and Q7**
> > > >
> > > > **Q1. The main concern about this paper is the experimental improvements are marginal.**
> > > >
> > > > **Q7. Can this proposed approach be applied to deeper and large Transformers such as BERT and GPTs and show its effectiveness?**
> > > >
> > > > Regarding the concern from the reviewer that our experimental improvements are marginal, in addition to Point (1) and (2) in our previous reply to your Q1,  we have run additional experiments to further verify that:
> > > >
> > > > (3) **The advantages of our Transformer-MGK/MLK models can be scaled up when applied on top of a larger and stronger baseline.**
> > > >
> > > > (4) **The advantages of our Transformer-MGK/MLK models still hold for different tasks including machine translation rather than just the LRA benchmark and Wikitext-103 language modeling.**
> > > >
> > > > Note that in all our experiments, we set **$M = 2$** for Transformer-MGK/MLK, i.e., using only 2 keys at each timestep.
> > > >
> > > > On Point (3), **as promised in our previous reply to your Q7**, in Table 3 in our revised manuscript, we have provided additional empirical results on WikiText-103 when applied our MGK/MLK approach on top of a larger and stronger baseline adapted from the medium model in [1]. This model has 90M parameters, 16 layers, 8 attention heads per layer, and a hidden size of 256. **The size of our baseline model is close to BERTBase** [2], which has 110M parameters, 12 layers, 12 attention heads per layer, and a hidden size of 768. Note that the baseline transformer we used is deeper than BERTBase. This baseline transformer attains a test perplexity (PPL) of 29.60 as reported in [1], which, on this WikiText-103 task, is better than or equivalent to popular transformer models including [3], [4], [5], [6], and [7]. Note that for [7], results on WikiText-103 can be found in Table 6 of [8]. **Applying our MGK on top of this new larger and stronger baseline, we significantly improve the test PPL to 28.86**.
> > > >
> > > > On Point (4), we have studied the advantage of the MGK over the baseline transformer **on the IWSLT14 German-English machine translation task. Compared to the baseline transformer model that uses 4-head softmax attention, our models that use MGK with only 2 heads achieve better or comparable BLEU scores (34.69 and 34.34 vs. 34.42 BLUE scores of the baseline)**. We have summarized our results in Section 3.3 in the main text and Table 9 in Appendix A.10 of the revised manuscript. We provide details on the dataset, model, and training in Appendix A.1.3. Again, using only 2 heads vs. 4 heads as in the baseline allows our model to save more parameters and achieve better computational efficiency.
> > > >
> > > > **References**
> > > >
> > > > [1] Imanol Schlag, Kazuki Irie, and Jurgen Schmidhuber. Linear Transformers are Secretly Fast Weight Programmers. ICML, 2021.
> > > >
> > > > [2] Jacob Devlin, Ming-Wei Chang, Kenton Lee, and Kristina Toutanova. BERT: Pre-training of deep bidirectional transformers for language understanding. NAACL, 2019.
> > > >
> > > > [3] Edouard Grave, Armand Joulin, and Nicolas Usunier. Improving neural language models with a continuous cache. ICLR, 2017.
> > > >
> > > > [4] Yann N. Dauphin, Angela Fan, Michael Auli, and David Grangier. Language modeling with gated convolutional networks. ICML, 2017.
> > > >
> > > > [5] Stephen Merity, Nitish Shirish Keskar, and Richard Socher. An analysis of neural language modeling at multiple scales. arXiv, abs/1803.08240, 2018.
> > > >
> > > > [6] Jack W. Rae, Chris Dyer, Peter Dayan, and Timothy P. Lillicrap. Fast parametric learning with activation memorization. ICML, 2018.
> > > >
> > > > [7] Peter Shaw, Jakob Uszkoreit, and Ashish Vaswani. Self-attention with relative position representations. NAACL, 2018.
> > > >
> > > > [8] Zihang Dai, Zhilin Yang, Yiming Yang, Jaime Carbonell, Quoc V Le, and Ruslan Salakhutdi- nov. Transformer-xl: Attentive language models beyond a fixed-length context. ACL, 2019.

---

> > > > > ### Author Response · Authors · 2021-11-25
> > > > > **Response to Reviewer qyjK - Any further questions on our current draft**
> > > > >
> > > > > We would like to thank you again for your reviews and feedback. We have updated our manuscript and added replies to your comments and questions with our latest experimental results above. We have also summarized the changes we made in the manuscript according to the suggestions from all reviewers in the Summary of the Revision near the top of the discussion.
> > > > >
> > > > > Given that your current score is 5, we would appreciate it if you could let us know if our responses have addressed your concerns and whether you still have any other questions on the current draft.
> > > > >
> > > > > We would be happy to do any follow-up discussion or address any additional comments.

---

> > ### Comment · Reviewer_qyjK · 2021-11-30
> > **Thanks for the response**
> >
> > Thanks for the authors' response, and most of my concerns and questions have been addressed.
> > With providing additional extensive experiment results, the efficacy of the proposed methods seems to be demonstrated.
> > I adjusted my score from 5 to 6.
> >
> > Thanks.

---

> > > ### Author Response · Authors · 2021-11-30
> > > **Thanks for your endorsement!**
> > >
> > > Thanks for your response and we appreciate your endorsement.

---

> ### Comment · Area_Chair_GjhM · 2021-11-25
> **Reminder to take a look at author responses**
>
> Dear reviewer, please take a look at the author's response and see if your concerns have been addressed. The authors suggest that there is a misunderstanding on comparisons to baselines.

---

> ### Author Response · Authors · 2021-11-29
> **Response to Reviewer qyjK - Any further questions before the deadline**
>
> We would like to thank you again for your reviews and feedback. We have updated our manuscript and added replies to your comments and questions with our latest experimental results above. We have also summarized the changes we made in the manuscript according to the suggestions from all reviewers in the Summary of the Revision near the top of the discussion.
>
> Given that your current score is 5 and the deadline is approaching, we would appreciate it if you could let us know if our responses have addressed your concerns and whether you still have any other questions on the current draft.
>
> We would be happy to do any follow-up discussion or address any additional comments.

---

### Official Review · Reviewer_gyah · 2021-11-02

**Correctness:** 3
**Technical Novelty And Significance:** 3
**Empirical Novelty And Significance:** 3
**Recommendation:** 8
**Confidence:** 4

**Main Review:**

Strong points

1. The proposed method is empirically efficient in terms of the number of parameters and computational cost in terms of FLOPS.

2. Although there is previous work for the RBF kernel interpretation of self-attention [1], the proposed probabilistic interpretation of self-attention provides an interesting direction for future research.

3. This paper provides experimental results on LRA benchmark and a language modeling benchmark to show the effectiveness of the proposed method.

4. The paper also introduces the linear-attention variant of the proposed method.

[1] Yao-Hung Hubert Tsai et al. Transformer Dissection: An Unified Understanding for Transformer’s Attention via the Lens of Kernel.  EMNLP-IJCNLP 2019

##########################################################################

Weak points

1.  Although the authors argue that Transformer-MGK is introduced to improve the explanation power of each key, increase the representation of each attention head, and reduce the chance of learning redundant heads, this paper provides only empirical results and no theoretical explanation for these improvements.
(1) It is valuable to describe the computational/space complexity of the proposed method and compare them with those of conventional methods.
(2) Although I can somehow expect that the mixture of Gaussian increases the representation power of keys, I cannot imagine why the proposed method reduces redundant heads and promote the diversity in attention pattern.

2. Since pre-training and finetuning paradigm is out of scope of LRA benchmark, I suggest the authors to conduct experiments for standard NLP benchmarks, such as GLUE, SQuAD, SuperGLUE, or Winogrande.




**Summary Of The Paper:**

The paper addresses the efficiency of Transformer models.
The paper provides a probabilistic view of self-attention in transformers and introduces a novel transformer model, Transformer-MGK, and its extensions.
In Transformer-MGK, each key of self-attention is modeled as a mixture of Gaussians whose mixture weights are estimated by the E-step of EM algorithm.
In the experiments, they show Transformer-MGK is comparable or better than Transformers on LRA benchmark and a language modeling benchmark.
In addition, the authors empirically show Transformer-MGK alleviates the head redundancy of conventional Transformer.

**Summary Of The Review:**

Overall, I vote for weak acceptance. I like the probabilistic interpretation of self-attention and the empirical results are strong.
My major concern is about the clarity of the paper (please see weak point #1 below).

---

> ### Author Response · Authors · 2021-11-15
> **Response to Reviewer gyah**
>
> Thank you for your thoughtful review and valuable feedback. Below we address your concerns.
>
> -----
>
> **Q1. Although the authors argue that Transformer-MGK is introduced to improve the explanation power of each key, increase the representation of each attention head, and reduce the chance of learning redundant heads, this paper provides only empirical results and no theoretical explanation for these improvements. (1) It is valuable to describe the computational/space complexity of the proposed method and compare them with those of conventional methods. (2) Although I can somehow expect that the mixture of Gaussian increases the representation power of keys, I cannot imagine why the proposed method reduces redundant heads and promote the diversity in attention pattern.**
>
> **Reply:** (1) As the reviewer suggests, we have included a theorem in Section 2.2 in the revised manuscript to show that our Transformer-MGKs/MLKs using a finite Gaussian mixture model (GMM) can approximate any distribution of the queries at each time step, i.e. $P(q_i|t_{j}=1)$. This is not possible if only a Gaussian distribution is used as in the baseline softmax transformers. Thus, the GMM used in Transformer-MGKs/MLKs at each time step can capture a wider range of the distribution of the queries. The proof of this theorem is provided in Appendix C.
>
> (2) We would like to thank you for your insightful comment. To simplify the discussion, assume that we use a Transformer-MGK with a mixture of two Gaussian distributions at each time step. Then, your question is equivalent to comparing three models: one model, named model 1, is with $H$ heads using the standard softmax Transformer; another model is with $H/ 2$ heads using the mixture of two Gaussian distributions for each timestep; and the remaining model is with $H$ heads using the mixture of two Gaussian distributions for each timestep. Empirically, Figures 2 (Left), 5 and 8 show that attention matrices of our models capture more diverse patterns than those of the baselines. Theoretically, to show that using MGK with a mixture of two Gaussian distributions at each time step promotes the diversity in attention pattern requires a metric to compare the diversities between different models. To find this metric is an interesting research direction and we will explore it in future work.
>
> -----
>
> **Q2. Since pre-training and finetuning paradigm is out of scope of LRA benchmark, I suggest the authors to conduct experiments for standard NLP benchmarks, such as GLUE, SQuAD, SuperGLUE, or Winogrande.**
>
> **Reply:**  We have launched additional experiments on the NLP benchmarks that the reviewer suggests. We will update those new results in our revised manuscript soon.
>
> -----
>
> We hope we have cleared your concerns about our work. We have also revised our manuscript according to your comments, and we would appreciate it if we can get your further feedback at your earliest convenience.

---

> > ### Comment · Reviewer_gyah · 2021-11-22
> > **Thank you for the response.**
> >
> > I have read everyone's comments and the author's response.
> > The authors' response has addressed my main concerns and I have increased my score from 6 to 8.
> >
> > A suggestion for the next version:
> > * Although the index of Gaussian components is subscript r, the number of components is M.    I would like to recommend using the same alphabet, i.e. r and R or m and M.

---

> > > ### Author Response · Authors · 2021-11-23
> > > **Thanks for your endorsement!**
> > >
> > > Thanks for your further feedback and we appreciate your endorsement. We will change the index of Gaussian components and the number of components to the same alphabet in the next version of our manuscript as you suggested.

---

> > > > ### Author Response · Authors · 2021-11-30
> > > > **Finetuning Results on GLUE tasks**
> > > >
> > > > Again, thanks for your feedback and endorsement.  As promised in our reply to your Q2 above, we have run additional experiments on tasks MNLI, QNLI, QQP, RTE, and STS-B in the GLUE benchmark to compare the finetuning performance of our 8-head BERT model that uses the MGK attention (BERT MGK) with the baseline 16-head BERT model that uses the softmax attention (BERT Softmax). We summarize our results in Table 1 below.
> > > >
> > > > Our 8-head BERT MGK outperforms the baseline 16-head BERT Softmax on QNLI, RTE, and STS-B and achieves a comparable result to that of the baseline on the MNLI and QQP task. Note that our 8-head BERT MGK is much more efficient than the baseline 16-head BERT Softmax. Our experiments follow the setting in [1]. Our reproduced results for the 16-head BERT Softmax model are worse than those reported in [1] might be due to some unexpected mismatch between our training setting and the setting in [1]. We will fix these mismatches and include full results on GLUE in the next version of our paper.
> > > >
> > > > **Table 2:** Performances of our 8-head BERT model that uses the MGK attention (BERT MGK) with the baseline 16-head BERT model that uses the softmax attention (BERT Softmax) on tasks MNLI, QNLI, QQP, RTE, and STS-B in the GLUE benchmark . On QNLI, RTE, and STS-B, our 8-head BERT MGK yields better results than the baseline 16-head BERT Softmax while on MNLI and QQP, our model and the baseline yield comparable results. Note that our 8-head BERT MGK is much more efficient than the baseline 16-head BERT Softmax.
> > > >
> > > > |       | MNLI (acc)    | QNLI (acc)    | QQP (F1)    | RTE (acc)    | STS-B (spearmanr)    |
> > > > | :---:        |    :----:   |     :----:   |     :----:   |     :----:   |    :----:   |
> > > > | 16-head BERT Softmax        |   **73.30**    |     81.16    |     **82.85**    |      66.06    |      85.09    |
> > > > | 8-head BERT MGK    |    72.74    |     **81.44**    |     82.48    |      **69.31**    |      **85.21**    |
> > > >
> > > > **References**
> > > >
> > > > [1] Peter Izsak, Moshe Berchansky, Omer Levy. How to Train {BERT} with an Academic Budget. EMNLP, 2021.

---

### Official Review · Reviewer_r2tu · 2021-11-02

**Correctness:** 3
**Technical Novelty And Significance:** 2
**Empirical Novelty And Significance:** 1
**Recommendation:** 5
**Confidence:** 5

**Main Review:**

Strengths
-------------

- The paper provides a nice probabilistic view on the attention matrix that could be a new tool to study the transformer architecture.
- Language modeling experiments besides LRA are greatly appreciated
- The paper is well written and easy to follow

Weaknesses
------------------

The most significant weakness is that the provided experiments do not provide adequate evidence regarding any benefit of the proposed method.
1. In both LRA and Wikitext-103 all scores way too close to be able to judge whether any of the methods performs better.
2. Even if they do perform better (the only one could be sMGK on LRA-Retrieval) the model is sufficiently changed that more tests would be required. For instance, is the performance improvement coming from using the distance instead of the dot product? Is it coming from the key shifting?
3. In all experiments, what is M? It is quite hard to compare the performance with different number of heads without having M for MGK and MLK.
4. Moreover, it would be interesting to see the performance of softmax and linear transformers with 4 heads on Wikitext-103. It is quite interesting in tables 1 and 2 that reducing the number of heads has minimal impact in performance.

Finally, it would be interesting and beneficial to the paper's main point to have a proposition regarding why parameterizing the attention matrix in this way would improve the diversity of the attention heads. The rank argument does not show diversity between attention heads given that the attention matrices can all be the same but have high rank. Moreover, when thinking about how diversity can be achieved with this formulation, it can also be measured and enforced with some type of loss or regularizer.

Suggestions
-----------------

- It would be much more informative to exchange FLOPS with wall-clock time and although number of parameters is important for large transformer models, so is the memory footprint in the GPU during training.

**Summary Of The Paper:**

The paper constructs a Gaussian mixture model with uniform prior and one Gaussian centered at each key and shows that the posterior distribution ie. the probability that a given query is generated by a given key matches the attention scores in the transformer architecture. In addition, it is proposed to describe each key as an additional Gaussian mixture and it is shown that the posterior is easy to compute and resembles the multi-head attention (given that the query remains the same for all heads). The paper proposes that this formulation can learn more diverse attention-heads and reduce the computation by replacing some heads with the Gaussian mixture heads which are easier to compute. Moreover, the authors show that it is straightforward to transfer the aforementioned formulation to linear transformers. Finally, the authors perform experiments on the LRA benchmark and language modelling experiments on Wikitext-103.

**Summary Of The Review:**

As mentioned in the top of the weaknesses section, the main reason for my recommendation is that there is not sufficient evidence showing an improvement in performance with the proposed method.

---

> ### Author Response · Authors · 2021-11-15
> **Response to Reviewer r2tu (1/3)**
>
> Thank you for your thoughtful review and valuable feedback. Below we address your concerns.
>
> -----
>
> **Q1. In both LRA and Wikitext-103 all scores way too close to be able to judge whether any of the methods performs better. The only one that could perform better than the baseline is sMGK on LRA-Retrieval.**
>
> **Reply:** We believe there is a misunderstanding of the advantage of our Transformer-MGK/MLK over the baselines. Please allow us to clear this misunderstanding by clarifying the key results from our experiments. In our experiments on LRA and Wikitext-103, we set **$M = 2$** for Transformer-MGK/MLK, i.e., using only 2 keys at each timestep, and show that:
>
> (1) **Transformer-MGK/MLK with half the number of heads is comparable or better than the baseline softmax and linear transformers with the full number of heads** while being more efficient in both computational cost and memory footprint.
>
> (2) **Using the same number of heads, Transformer-MGK/MLK significantly outperforms the baseline softmax and linear transformers.**
>
> On Point (1), Tables 1, 2, and 3 in our paper verify that our Transformer-MGKs/MLKs with 1 head, 2 heads, and 4 heads outperform or at least on par with the baseline softmax and linear transformers with double the number of heads, i.e., 2 heads, 4 heads, and 8 heads, respectively, on LRA and Wikitext-103. It is interesting to notice from Tables 1 and 2 that even our 2-head Transformer-MGK/MLK models achieve better or equivalent results to the 12-head and 8-head baselines on the LRA benchmark. Using only half the number of heads or even fewer than that, our Transformer-MGKs/MLKs need much fewer parameters and FLOPs to compute than the baseline models. We believe this advantage of our models is significant.
>
> On Point (2), Tables 1, 2, and 3 confirm that our Transformer-MGKs/MLKs with the same number of heads as the baseline models outperform the baselines. In particular, on the task that requires very long-range attention such as the retrieval task, our Transformer-MGKs/MLKs significantly outperform the baselines.  Note that this retrieval task has the longest average sequence length and attention span among the LRA tasks (Tay, Yi, et al. Long range arena: A benchmark for efficient transformers. ICLR 2021). Also, on the Wikitext-103 task, our Transformer-MGKs/MLKs with 4 heads significantly outperform the baselines with 4 heads. The advantage gaps between our Transformer-MGKs/MLKs with 8 heads and the baselines with 8 heads on the Wikitext-103 task are smaller because the baselines with 8 heads, given their capacity, already perform very well on this Wikitext-103 task. That says, the improvement in accuracy gained by Transformer-MGKs/MLKs in this case is indeed relevant.

---

> > ### Author Response · Authors · 2021-11-15
> > **Response to Reviewer r2tu (2/3)**
> >
> > **Q2. Even if they do perform better (the only one could be sMGK on LRA-Retrieval) the model is sufficiently changed that more tests would be required. For instance, is the performance improvement coming from using the distance instead of the dot product? Is it coming from the key shifting?**
> >
> > **Reply:** First, we respectfully disagree with the reviewer’s comment that the only one among our Transformer-MGK/MLK models that do perform better than the baseline is sMGK on the LRA Retrieval task and provide a detailed explanation in our Reply to your Q1 above.
> >
> > Second, as the reviewer suggests, we have done an ablation study of the Transformer-MGK on the LRA retrieval task to investigate where the performance improvement is from. In particular, we would like to understand the impact of the following factors on the performance of Transformer-MGK: 1) the mixture of keys, 2) the Gaussian distance, and 3) the key shifting. We summarize our empirical results in Table 1 below and discuss the impact of 1, 2 and 3 below. We have included this ablation study in Appendix A.4  in our revised manuscript.
> >
> >
> > 1) **Impact of the Mixture of Keys:** We apply our mixture of keys (MGK) approach to the softmax transformer using the dot product between queries and keys instead of the Gaussian distance as in our paper. We name this model Softmax MGK. We compare the Softmax MGK that has 1 head (Softmax MGK 1 head in Table 1 below) with the baseline softmax transformers that use 1 and 2 heads (Softmax 1 head and Softmax 2 heads in Table 1 below). Results in Table 1 show that the Softmax MGK  1 head outperforms both the baseline softmax transformers of 1 and 2 heads. Note that our Softmax MGK 1 head is more efficient than the baseline of 2 heads in terms of the number of parameters and the number of FLOPs. These results confirm the benefit of using MGK.
> >
> >
> > 2) **Impact of using the Gaussian distance:** Next, we compare the Softmax MGK with the Gaussian MGK. Here the Gaussian MGK is the Transformer-MGK proposed and discussed in our paper, which computes the attention scores using the MGK approach and the Gaussian distance between the queries and keys. Results in Table 1 below suggest that the Gaussian MGK 1 head improves over the Softmax MGK 1 head (80.63% vs. 79.23%). This result justifies the advantage of using Gaussian distance over dot product to compute the attention scores.
> >
> >
> > 3) **Impact of key shifting:** Finally, we apply key shifting to both Softmax MGK and Gaussian MGK (Softmax sMGK and Gaussian sMGK in Table 1 below). From Table 1, we observe that the softmax sMGK 1 head and Gaussian sMGK 1 head outperform the softmax MGK 1 head and Gaussian MGK 1 head, respectively. These results, again, corroborate the benefit of using key shifting.
> >
> >
> > We also include the result for the Gaussian 1 head model in Table 1. This Gaussian 1 head model is similar to the softmax 1 head model but uses the Gaussian distance to compute the attention scores. Comparing the results of the Gaussian 1 head model, the softmax 1 head model, and the Gaussian MGK 1 head model reported in Table 1 further confirms the advantage of using MGK and Gaussian distance.
> >
> >
> > **Table 1:** Ablation study on the impact of 1) the mixture of keys, 2) the Gaussian distance, and 3) the key shifting on the LRA retrieval task.  We denote the softmax transformer by Softmax. All Softmax models (i.e. Softmax 2 heads, Softmax 1 head, Softmax MGK 1 head, and Softmax sMGK 1 head) use dot product to compute the attention scores. All Gaussian models (i.e. Gaussian MGK 1 head, Gaussian sMGK 1 head, and Gaussian 1 head) use Gaussian distance to compute the attention scores. We denote MGK with key shifting by sMGK. Here MGK is used to denote our approach of using a mixture of keys at each timestep.
> >
> >
> > | Method       | Accuracy    |
> > | :---        |    :----:   |
> > | *Softmax 2 heads*       |    79.10    |
> > | Softmax sMGK 1 head    |    79.81    |
> > | Softmax MGK 1 head    |    79.23   |
> > | *Softmax 1 head*    |    77.90    |
> > | | |
> > | Gaussian sMGK 1 head    |    81.23    |
> > | Gaussian MGK 1 head    |    80.63   |
> > | *Gaussian 1 head*    |    80.38    |

---

> > > ### Author Response · Authors · 2021-11-15
> > > **Response to Reviewer r2tu (3/3)**
> > >
> > > **Q3. In all experiments, what is M? It is quite hard to compare the performance with different number of heads without having M for MGK and MLK.**
> > >
> > > **Reply:** In all experiments, we set $M = 2$. Then we compare the baseline softmax and linear transformers with our Transformer-MGKs/MLKs ($M=2$) that use half the number of heads and the full number of heads. We’ve made this clear in the second paragraph of section 3 in the revised manuscript.
> > >
> > > -----
> > >
> > > **Q4. Moreover, it would be interesting to see the performance of softmax and linear transformers with 4 heads on Wikitext-103. It is quite interesting in tables 1 and 2 that reducing the number of heads has minimal impact in performance.**
> > >
> > > **Reply:** We have included results for softmax and linear transformers with 4 heads on WikiText-103 in Table 3 in the revised manuscript. For these baseline models, we see that reducing the number of heads from 8 to 4 significantly decreases the performance of the models with more than 1.5 reduction in validation and test PPL for the softmax transformer and more than 1.0 reduction in validation and test PPL for the linear transformer. Our proposed Transformer-MGK and Transformer-MLK help close this gap. In particular, Transformer-MGK with 4 heads outperforms the baseline softmax transformer with 8 heads in test PPL, and Transformer-MLK with 4 heads performs on par with the baseline linear transformer with 8 heads. We have made these points clear in our revision.
> > >
> > > -----
> > >
> > > **Q5. Finally, it would be interesting and beneficial to the paper's main point to have a proposition regarding why parameterizing the attention matrix in this way would improve the diversity of the attention heads. The rank argument does not show diversity between attention heads given that the attention matrices can all be the same but have high rank.**
> > >
> > > **Reply:** As the reviewer suggests, we have included a theorem in Section 2.2 in the revised manuscript to show that our Transformer-MGKs/MLKs using a finite Gaussian mixture model can sufficiently approximate well any distribution of the queries at each time step, i.e. $P(q_i|t_{j}=1)$. This is not possible if only a Gaussian distribution is used as in the baseline softmax transformers. Thus, the GMM used in Transformer-MGKs/MLKs at each time step can capture a wider range of the distribution of the queries. The proof of the theorem is provided in Appendix C.
> > >
> > > Theoretically, to show that using MGK with a mixture of Gaussian distributions at each time step promotes the diversity in attention pattern requires a metric to compare the diversities between different models. To find this metric is an interesting research direction and we will explore it in future work. Empirically, Figures 2 (Left), 5 and 8 show that attention matrices of our Transformer-MGK/MLK models capture more diverse patterns than those of the baselines.
> > >
> > > -----
> > >
> > > **Q6. Moreover, when thinking about how diversity can be achieved with this formulation, it can also be measured and enforced with some type of loss or regularizer.**
> > >
> > > **Reply:** Thanks for your suggestion. One way to qualitatively measure this diversity is to visualize the attention matrices as we do in Figures 2, 5, and 8 in our paper. As mentioned in our Reply to your Q5 above, we believe that finding a reliable quantitative metric to measure the diversity of attention patterns in transformers is a relevant research problem for future work.
> > >
> > > -----
> > >
> > > **Q7. It would be much more informative to exchange FLOPS with wall-clock time and although number of parameters is important for large transformer models, so is the memory footprint in the GPU during training.**
> > >
> > > **Reply:** As you suggested, we have run additional experiments to compare the time overhead (wall-clock time) of our models with the baseline on the retrieval task in the LRA benchmark. We have also compared the memory footprint in the GPU during training on the same task. Our Transformer-MGK/MLK is more efficient than the corresponding baseline softmax and linear transformers in both metrics. We have included these new results in Table 6 in Appendix A.5 of the revised manuscript.
> > >
> > > -----
> > >
> > > We hope we have cleared your concerns about our work. We have also revised our manuscript according to your comments, and we would appreciate it if we can get your further feedback at your earliest convenience.

---

> > > > ### Author Response · Authors · 2021-11-21
> > > > **Response to Reviewer r2tu - New Empirical Results to Further Address Q1**
> > > >
> > > > **Q1. In both LRA and Wikitext-103 all scores way too close to be able to judge whether any of the methods performs better. The only one that could perform better than the baseline is sMGK on LRA-Retrieval.**
> > > >
> > > > **Reply:** Regarding the concern from the reviewer that there is not sufficient evidence showing an improvement in performance with our proposed Transformer with Mixture of Gaussian Keys model (Transformer-MGK), in addition to Point (1) and (2) in our previous reply to this question,  we have run additional experiments to further verify that:
> > > >
> > > > (3) **The advantages of our Transformer-MGK/MLK models can be scaled up when applied on top of a larger and stronger baseline.**
> > > >
> > > > (4) **The advantages of our Transformer-MGK/MLK models still hold for different tasks including machine translation rather than just the LRA benchmark and Wikitext-103 language modeling.**
> > > >
> > > > Note that in all our experiments, we set **$M = 2$** for Transformer-MGK/MLK, i.e., using only 2 keys at each timestep.
> > > >
> > > > On Point (3), in Table 3 in our revised manuscript, we have provided additional empirical results on WikiText-103 when applied our MGK/MLK approach on top of a larger and stronger baseline adapted from the medium model in [1]. This model has 90M parameters, 16 layers, 8 attention heads per layer, and a hidden size of 256. **The size of our baseline model is close to BERTBase** [2], which has 110M parameters, 12 layers, 12 attention heads per layer, and a hidden size of 768. Note that the baseline transformer we used is deeper than BERTBase. This baseline transformer attains a test perplexity (PPL) of 29.60 as reported in [1], which, on this WikiText-103 task, is better than or equivalent to popular transformer models including [3], [4], [5], [6], and [7]. Note that for [7], results on WikiText-103 can be found in Table 6 of [8]. **Applying our MGK on top of this new larger and stronger baseline, we significantly improve the test PPL to 28.86**.
> > > >
> > > > On Point (4), we have studied the advantage of the MGK over the baseline transformer **on the IWSLT14 German-English machine translation task. Compared to the baseline transformer model that uses 4-head softmax attention, our models that use MGK with only 2 heads achieve better or comparable BLEU scores (34.69 and 34.34 vs. 34.42 BLUE scores of the baseline)**. We have summarized our results in Section 3.3 in the main text and Table 9 in Appendix A.10 of the revised manuscript. We provide details on the dataset, model, and training in Appendix A.1.3. Again, using only 2 heads vs. 4 heads as in the baseline allows our model to save more parameters and achieve better computational efficiency.
> > > >
> > > > **References**
> > > >
> > > > [1] Imanol Schlag, Kazuki Irie, and Jurgen Schmidhuber. Linear Transformers are Secretly Fast Weight Programmers. ICML, 2021.
> > > >
> > > > [2] Jacob Devlin, Ming-Wei Chang, Kenton Lee, and Kristina Toutanova. BERT: Pre-training of deep bidirectional transformers for language understanding. NAACL, 2019.
> > > >
> > > > [3] Edouard Grave, Armand Joulin, and Nicolas Usunier. Improving neural language models with a continuous cache. ICLR, 2017.
> > > >
> > > > [4] Yann N. Dauphin, Angela Fan, Michael Auli, and David Grangier. Language modeling with gated convolutional networks. ICML, 2017.
> > > >
> > > > [5] Stephen Merity, Nitish Shirish Keskar, and Richard Socher. An analysis of neural language modeling at multiple scales. arXiv, abs/1803.08240, 2018.
> > > >
> > > > [6] Jack W. Rae, Chris Dyer, Peter Dayan, and Timothy P. Lillicrap. Fast parametric learning with activation memorization. ICML, 2018.
> > > >
> > > > [7] Peter Shaw, Jakob Uszkoreit, and Ashish Vaswani. Self-attention with relative position representations. NAACL, 2018.
> > > >
> > > > [8] Zihang Dai, Zhilin Yang, Yiming Yang, Jaime Carbonell, Quoc V Le, and Ruslan Salakhutdi- nov. Transformer-xl: Attentive language models beyond a fixed-length context. ACL, 2019.

---

> > > > > ### Author Response · Authors · 2021-11-25
> > > > > **Response to Reviewer r2tu - Any further questions on our current draft**
> > > > >
> > > > > We would like to thank you again for your reviews and feedback. We have updated our manuscript and added replies to your comments and questions with our latest experimental results above. We have also summarized the changes we made in the manuscript according to the suggestions from all reviewers in the Summary of the Revision near the top of the discussion.
> > > > >
> > > > > Given that your current score is 5, we would appreciate it if you could let us know if our responses have addressed your concerns and whether you still have any other questions on the current draft.
> > > > >
> > > > > We would be happy to do any follow-up discussion or address any additional comments.

---

> ### Comment · Area_Chair_GjhM · 2021-11-25
> **reminder**
>
> Dear reviewer, please take a look at the author's response and see if your concerns have been addressed. The authors suggest that there is a misunderstanding on comparisons to baselines. Do you agree?

---

> ### Author Response · Authors · 2021-11-29
> **Response to Reviewer r2tu - Any further questions before the deadline**
>
> We would like to thank you again for your reviews and feedback. We have updated our manuscript and added replies to your comments and questions with our latest experimental results above. We have also summarized the changes we made in the manuscript according to the suggestions from all reviewers in the Summary of the Revision near the top of the discussion.
>
> Given that your current score is 5 and the deadline is approaching, we would appreciate it if you could let us know if our responses have addressed your concerns and whether you still have any other questions on the current draft.
>
> We would be happy to do any follow-up discussion or address any additional comments.

---

### Official Review · Reviewer_EtfP · 2021-11-03

**Correctness:** 3
**Technical Novelty And Significance:** 3
**Empirical Novelty And Significance:** 2
**Recommendation:** 8
**Confidence:** 4

**Main Review:**


- First, I like the Gaussian mixture perspective of multihead attention in general, but it could lead to cases that can be a bit awkward. For example, in causal attention, the number of components of the mixture changes over time steps. This could be avoided by, e.g., treating only the multihead part as a mixture, not the keys over timesteps.
- Second, it can be confusing to weave into the narrative something that is actually not used at all, e.g., the norm constraints around Eq 5, and the M step around Eq 13. I suggest reworking these part and probably presenting them as connections to other models.
- Third, some modeling details need to be clarified. Optional (A) for key’s parameterization, the notation W_{k_{jr}} seems to suggest it needs NM number of projection matrices. This seems  much more expensive than standard multihead attention. Similarly in option (B), why would it need N number of matrices at all? Further about the parameterizations of queries and values. The notations seem to suggest that they are never “multiheaded” as keys are. Could the authors confirm this? If this is true, it is worth clarifying.
- Adding onto the above, what does a MGK with 4 heads mean? Does it have 4 x seq_len number of Gaussian components for the keys?
- Additionally, it would be nice to include a time overhead comparison.
- Lastly about baselines and further experiments: the WikiText103 baseline seems pretty far behind, e.g., Baeski and Auli (2018). Could the authors explore MGK on top of a stronger baseline? Besides, [1] seems pretty related and worth comparing to. Given the weak LM baseline, I suggest the authors including other sequence modeling tasks such as machine translation.

Typos:
- Above 1.1, keyed -> key.
- Above Eq 4, unit matrix -> identity
- Above Eq 7, abusing -> overloading
- Below Eq 11, at learning -> at learning time.
- Start of Section 3, “full the number of heads” doesn’t parse for me.

References:

[1] https://arxiv.org/abs/1911.02150

[2] https://arxiv.org/abs/2103.02143

**Summary Of The Paper:**

This paper proposes a mixture-model perspective of multihead attention. Each key within each attention head is treated as a Gaussian component; with some assumptions, the softmax-based attention weights can be recovered as the posterior probability of the Gaussian mixture, conditioning on the query. The framework leads to a new parameterization of multihead attention, which is explored in transformers. It can be applied to several linear transformer variants, too. Experiments with text classification, language modeling show that the proposed method achieves same or better accuracy controlling the number of attention heads, and reduces computational overhead. Analysis suggests that it reduces the redundancy of multihead attention

**Summary Of The Review:**

Strengths:
- An interesting mixture model perspective of multihead attention
- It is nice to reduce compute cost with the same or better accuracy

Weaknesses:
- It can be confusing to introduce concepts that are never used in the model
- Several key details of the implementation need to be clarified.
- Weak LM baseline. Would be great to include more experiments.

---

> ### Author Response · Authors · 2021-11-15
> **Response to Reviewer EtfP (1/2)**
>
> Thank you for your thoughtful review and valuable feedback. Below we address your concerns.
>
> -----
>
> **Q1. First, I like the Gaussian mixture perspective of multihead attention in general, but it could lead to cases that can be a bit awkward. For example, in causal attention, the number of components of the mixture changes over time steps. This could be avoided by, e.g., treating only the multihead part as a mixture, not the keys over timesteps.**
>
> **Reply:** Thanks for your interesting suggestion. As mentioned in Section 2.1, we set the prior $\pi_{j}$ of the keys at different timesteps in Eq. 4 to be uniform. This helps bypass the modeling challenge when the number of components of the mixture changes over time steps as in causal attention. We will consider learning the prior $\pi_{j}$ and treating only the multihead part as a mixture as the reviewer suggests in our future work. Note that, in our Mixture of Gaussian Keys and Mixture of Linear Keys model, in Eq. 7, we still learn the prior distribution $\pi_{jr}$ of the keys in the same time step.
>
> -----
>
> **Q2. Second, it can be confusing to weave into the narrative something that is actually not used at all, e.g., the norm constraints around Eq 5, and the M step around Eq 13. I suggest reworking these part and probably presenting them as connections to other models.**
>
> **Reply:** Thanks for your suggestion. The norm constraints, i.e., the assumption that the query $q_{i}$ and the key $k_{j}$ are normalized, are needed to derive the correspondence between the posterior $p(t_{j}=1|q_{i})$ of our Gaussian mixture model and the attention score in self-attention in Eq. 6. We agree with the reviewer that the details of the M step around Eq. 13 can be removed for more concise and clearer writing. We have revised our manuscript according to your suggestion, and this indeed helps improve the readability of our paper significantly.
>
> -----
>
> **Q3. Third, some modeling details need to be clarified. Optional (A) for key’s parameterization, the notation W_{k_{jr}} seems to suggest it needs NM number of projection matrices. This seems much more expensive than standard multihead attention. Similarly in option (B), why would it need N number of matrices at all? Further about the parameterizations of queries and values. The notations seem to suggest that they are never “multiheaded” as keys are. Could the authors confirm this? If this is true, it is worth clarifying. Adding onto the above, what does a MGK with 4 heads mean? Does it have 4 x seq_len number of Gaussian components for the keys?**
>
> **Reply:** Thanks for your comments. Those are indeed our typos. Optional (A) for key’s parameterization should be $k_{jr} = x_{j} W_{K_{r}^{\top}}$ instead, where $x_{j} \in \mathbb{R}^{1\times D_x}$, $W_{K_{r}}\in \mathbb{R}^{D\times D_{x}}$ and $r=1,2, \dots, M$. Thus, it only needs $M$ projection matrices to compute the keys in each head. Note that in our experiments, we set $M=2$ and propose to reduce the number of heads by half. As a result, our model needs the same projection matrices to compute the keys as the baseline multihead attention with the full number of heads while only needing half the number of projection matrices to compute the query and value vectors. Similarly, in option (B), $k_{jr} = x_{j} W_{K}^{\top} + b_{r}$, and it only need one projection matrix to compute the keys in each head. In the revised manuscript, we have corrected these typos in the “Design Options for Keys” paragraph in section 2.3 and the 2nd paragraph in section 3. A more detailed comparison of model complexity and computational cost between the Transformer-MGK and the baseline softmax transformer is given in Appendix B.
>
> The queries and values are also “multiheaded” as the keys are. In particular, for each head of the baseline transformers and our models, there is a set projection matrices of $\{W_{Q}, W_{K}, W_{V}\}$. Those matrices in different heads are different. Thus the queries $Q$, the keys $K$, and the value $V$ are different at different heads.
>
> In an MGK with 4 heads, each head uses a GMM to model the distribution of the queries $q_{i}$ as in Eq. 4. In each head, at each timestep, a GMM with $M$ components is used to model the distribution $P(q_{i}|t_{j}=1)$ as in Eq. 7, which leads to $M$ keys at each timestep. Therefore, an MGK with 4 heads has 4 x M x seq_len number of Gaussian components for the keys. In our experiment, we set $M=2$ and compare our MGK models with the 8-head softmax transformers.

---

> > ### Author Response · Authors · 2021-11-15
> > **Response to Reviewer EtfP (2/2)**
> >
> > **Q4. Additionally, it would be nice to include a time overhead comparison.**
> >
> > **Reply:** As you suggested, we have run additional experiments to compare the time overhead of our models with the baseline on the retrieval task in the LRA benchmark. We have also compared the memory footprint in the GPU during training on the same task. Our Transformer-MGK/MLK is more efficient than the corresponding baseline softmax and linear transformers in both metrics. We have included these new results in Table 6 in Appendix A.5 of the revised manuscript.
> >
> > -----
> >
> > **Q5. Lastly about baselines and further experiments: the WikiText103 baseline seems pretty far behind, e.g., Baeski and Auli (2018). Could the authors explore MGK on top of a stronger baseline? Besides, [1] seems pretty related and worth comparing to. Given the weak LM baseline, I suggest the authors including other sequence modeling tasks such as machine translation.
> > Baeski and Auli (2018): https://arxiv.org/pdf/1809.10853.pdf
> > [1] Rami Al-Rfou, DK Choe, Noah Constant, Mandy Guo, and Llion Jones. Character-level language modeling with deeper self-attention. In Thirty-Third AAAI Conference on Artificial Intelligence, 2019a. URL https://arxiv.org/abs/1808.04444.**
> >
> > **Reply:** We have launched experiments that use MGK on top of a stronger baseline for the WikiText-103 to compare with the proposed models in [1]. We have also been working on additional experiments on machine translation tasks. We will update those new results in our revised manuscript soon. Both language modeling on WikiText-103 and machine translation tasks take a few days on the machine in our lab.
> >
> > -----
> >
> > **Q6. Above 1.1, keyed -> key, Above Eq 4, unit matrix -> identity, Above Eq 7, abusing -> overloading, Below Eq 11, at learning -> at learning time, Start of Section 3, “full the number of heads” doesn’t parse for me.**
> >
> > **Reply:** Thank you so much for pointing these out. We have addressed them in our revision.
> >
> > -----
> >
> > We hope we have cleared your concerns about our work. We have also revised our manuscript according to your comments, and we would appreciate it if we can get your further feedback at your earliest convenience.

---

> > > ### Author Response · Authors · 2021-11-21
> > > **Response to Reviewer EtfP - New Empirical Results to Address Q5**
> > >
> > > **Q5. Lastly about baselines and further experiments: the WikiText103 baseline seems pretty far behind, e.g., Baeski and Auli (2018). Could the authors explore MGK on top of a stronger baseline? Besides, [1] seems pretty related and worth comparing to. Given the weak LM baseline, I suggest the authors including other sequence modeling tasks such as machine translation. Baeski and Auli (2018): https://arxiv.org/pdf/1809.10853.pdf [1] Rami Al-Rfou, DK Choe, Noah Constant, Mandy Guo, and Llion Jones. Character-level language modeling with deeper self-attention. In Thirty-Third AAAI Conference on Artificial Intelligence, 2019a. URL https://arxiv.org/abs/1808.04444.**
> > >
> > > **Reply:**
> > > We would like to thank the reviewer for the comments. To address the reviewer’s first suggestion about exploring MGK on top of a stronger baseline, we have launched additional experiments on WikiText-103 with a larger baseline model adapted from the medium model in [2]. This model has 90M parameters, 16 layers, 8 attention heads per layer, and hidden size of 256. **The size of our baseline model is close to BERTBase** [3], which has 110M parameters, 12 layers, 12 attention heads per layer, and hidden size of 768. Note that the baseline transformer we used is deeper than BERTBase. This baseline transformer attains a test perplexity (PPL) of 29.60 as reported in [2], which is better than or equivalent to [Grave et. al. (2016)], [Dauphin et al. (2017)], [Merity et al. (2018)], and [Rae et al. (2018)] reported in Table 2 in the paper [Baeski and Auli (2018)] that the reviewer mentioned. Our baseline transformer also attains better test PPL than (Shaw et al. (2018)). Note that for (Shaw et al. (2018), results on WikiText-103 can be found in Table 6 of Dai et al. (2019)). **Applying our MGK on top of this baseline, we significantly improve the test PPL from 29.60 to 28.86**. We have updated Table 3 in our revised manuscript to include these new results.
> > >
> > > To address the reviewer’s second suggestion about comparing to [1], we will first summarize the key contribution of [1] here. This paper proposes to use a deeper transformer architecture of 64 transformer layers trained with additional auxiliary losses. Those losses include prediction losses for each position in the sequence, intermediate losses at each layer in the network, and losses from multiple targets, which make two or more predictions of future characters. Learned per-layer positional embedding instead of fixed sinusoid timing signal is also used to facilitate the training of such a deep network.  These techniques are complementary to our MGK, which models each key in self-attention in transformers as a Gaussian mixture model, and can be incorporated into our MGK to further improve its performance. We have added a comparison to [1] and also to [Baeski and Auli (2018)] in the Related Work section of our revision.
> > >
> > > Finally, according to the reviewer’s suggestion, we have studied the advantage of the MGK over the baseline transformer **on the IWSLT14 De-En machine translation task. Compared to the baseline transformer model that uses 4-head softmax attention, our model that uses MGK with only 2 heads achieves better or comparable BLEU scores (34.69 and 34.34 vs. 34.42 BLUE score of the baseline)**. We have summarized our results in section 3.3 in the main text and Table 9 in Appendix A.10 of the revised manuscript. We provide details on the dataset, model, and training in Appendix A.1.3. Again, using only 2 heads vs 4 heads as in the baseline allows our model to save more parameters and achieve better computational efficiency.
> > >
> > > **References**
> > >
> > > [1] Rami Al-Rfou, DK Choe, Noah Constant, Mandy Guo, and Llion Jones. Character-level language modeling with deeper self-attention. AAAI, 2019.
> > >
> > > [2] Imanol Schlag, Kazuki Irie, and Jurgen Schmidhuber. Linear Transformers are Secretly Fast Weight Programmers. ICML, 2021.
> > >
> > > [​​3] Jacob Devlin, Ming-Wei Chang, Kenton Lee, and Kristina Toutanova. BERT: Pre-training of Deep Bidirectional Transformers for Language Modeling. NAACL, 2019.
> > >
> > > [4] Edouard Grave, Armand Joulin, and Nicolas Usunier. Improving neural language models with a continuous cache. ICLR, 2017.
> > >
> > > [5] Yann N. Dauphin, Angela Fan, Michael Auli, and David Grangier. Language modeling with gated convolutional networks. ICML, 2017.
> > >
> > > [6] Stephen Merity, Nitish Shirish Keskar, and Richard Socher. An analysis of neural language modeling at multiple scales. arXiv, abs/1803.08240, 2018.
> > >
> > > [7] Jack W. Rae, Chris Dyer, Peter Dayan, and Timothy P. Lillicrap. Fast parametric learning with activation memorization. ICML, 2018.
> > >
> > > [8] Peter Shaw, Jakob Uszkoreit, and Ashish Vaswani. Self-attention with relative position representations. NAACL, 2018.
> > >
> > > [9] Zihang Dai, Zhilin Yang, Yiming Yang, Jaime Carbonell, Quoc V Le, and Ruslan Salakhutdi- nov. Transformer-xl: Attentive language models beyond a fixed-length context. ACL, 2019.

---

> > > > ### Comment · Reviewer_EtfP · 2021-11-22
> > > > **Thanks for the response!**
> > > >
> > > > The authors' response has addressed most of my concerns, which I appreciate. A remaining one that is completely on me: by suggesting comparing to [1] I meant this paper: https://arxiv.org/abs/1911.02150. I forgot to paste the reference section into my original review. Sorry.
> > > >
> > > > The suggested baseline uses a single attention head for keys and values, which shares some idea with this paper. At this point, the authors won't have enough time to compare to it. Since this was my fault, it won't be factored into my score. But I hope the authors can include this comparison in the next version.
> > > >
> > > > I have adjusted my score accordingly. Below is the reference section I wanted to put into the original review:
> > > >
> > > > References:
> > > >
> > > > [1] https://arxiv.org/abs/1911.02150
> > > >
> > > > [2] https://arxiv.org/abs/2103.02143

---

> > > > > ### Author Response · Authors · 2021-11-23
> > > > > **Thanks for your endorsement!**
> > > > >
> > > > > Thanks for your further feedback and we appreciate your endorsement. We have studied the multi-query attention in [1] and provided a comparison between our Transformer-MGK and this method below.  We have included this comparison in Appendix A.11 and added [1] into the Related Work section of our revised manuscript as you suggested. We will run additional experiments to compare the multi-query attention and Transformer-MGK and include those new results in the next version of our manuscript. We have also cited [2], which uses random feature methods to approximate the softmax attention and is a relevant work in efficient transformers, in the Related Work section of our revision.
> > > > >
> > > > > **Comparison between the multi-query attention and our MGK approach:** The multi-query attention shares the same set of keys and values at different heads to reduce the memory-bandwidth cost during incremental inference, which allows faster inference since the size of the reloaded "keys" and "values" tensors are significantly reduced. On another hand, our Transformer-MGK models the key at each head as a Gaussian mixture model, which leads to the use of multiple keys at each head and allows us to decrease the number of attention heads. This helps reduce the computations and parameters needed to calculate additional queries and values. If using key shifting (see option (B) in paragraph Design Options for Keys in section 2.3 of our manuscript), the MGK approach also helps reduce the computations and parameters needed to calculate additional keys. The advantages of Transformer-MGK hold in both training and inference, including incremental inference. We have provided a detailed analysis of the computational complexity and the number of parameters of the Transformer-MGK in comparison with the corresponding softmax transformer in Appendix B of our revised manuscript. Combining the multi-query attention and our MGK approach is interesting since each method has its own advantage and they are complementary to each other. In particular, we can let the transformer model share the same set of values and mixtures of keys at different heads. This approach can potentially have the advantages of both multi-query attention and MGK.

---

> > > > > > ### Author Response · Authors · 2021-11-30
> > > > > > **Empirical Results for Comparing Multi-query and Transformer MGK**
> > > > > >
> > > > > > Again, thanks for your feedback and endorsement. As promised, we have run additional experiments on the Wikitext-103 language modeling task to compare multi-query attention [1] and Transformer-MGK. Our experiments follow the small model setting from [2]. We summarize those new empirical results in Table 1 below and will include them in the next version of our paper. The 4-head Transformer-MGK yields better test PPL than the baseline 8-head softmax transformers and significantly outperforms the 8-head transformer with multi-query attention. Incorporating multi-query attention into the 4-head transformer-MGK still yields better validation and test PPL than the 8-head multi-query. This 4-head transformer-MGK with multi-query attention inherits the advantages in efficiency from both multi-query attention and transformer-MGK. Compared to the 4-head transformer-MGK, it has a much smaller memory-bandwidth cost during incremental inference. Compared to the baseline softmax transformer and the transformer with multi-query attention, it is more efficient in terms of model complexity and computational cost.
> > > > > >
> > > > > > **Table 1:** Perplexity (PPL) on Wikitext-103 of 4-head Transformer-MGK with and without using multi-query attention in comparison with the baseline 8-head softmax transformer with and without using multi-query attention. The 4-head Transformer-MGK yields better test PPL than the baseline 8-head softmax transformers and significantly outperforms the 8-head transformer with multi-query attention. The 4-head transformer-MGK with multi-query attention yields better results than the 8-head transformer with multi-query attention while inheriting the advantages in efficiency from both multi-query attention and transformer-MGK.
> > > > > >
> > > > > >
> > > > > > | Method       | Valid PPL    | Test PPL    |
> > > > > > | :---        |    :----:   |     :----:   |
> > > > > > | Softmax 8 heads       |    33.15    |     34.29    |
> > > > > > | Multi-query 8 heads    |    35.08    |     36.03    |
> > > > > > | Transformer-MGK 4 head    |    33.28   |     **34.21**    |
> > > > > > | Multi-query + Transformer-MGK 4 head    |    35.16    |    **35.79**    |
> > > > > >
> > > > > > **References**
> > > > > >
> > > > > > [1] Noam  Shazeer.    Fast  transformer  decoding:   One  write-head  is  all  you  need. arXiv  preprint arXiv:1911.02150, 2019.
> > > > > >
> > > > > > [2] Imanol Schlag, Kazuki Irie, and Jurgen Schmidhuber. Linear Transformers are Secretly Fast Weight Programmers. ICML, 2021.

---

### Author Response · Authors · 2021-11-15
**General Response**

Dear AC and reviewers,

Thanks for your thoughtful reviews and valuable comments, which have helped us improve the paper significantly. We updated our submission based on the reviewers' feedback, and we have highlighted our revision in blue.

One of the common comments is that there is not sufficient evidence showing an improvement in performance with our proposed Transformer with a Mixture of Gaussian Keys model (Transformer-MGK). We first address this comment here.

**Empirically**, in our experiments on LRA and Wikitext-103, we set **$M = 2$** for Transformer-MGK/MLK, i.e., using only 2 keys at each timestep, and show that:

(1) **Transformer-MGK/MLK with half the number of heads is comparable or better than the baseline softmax and linear transformers with the full number of heads** while being more efficient in both computational cost and memory footprint.

(2) **Using the same number of heads, Transformer-MGK/MLK significantly outperforms the baseline softmax and linear transformers.**

On Point (1), Tables 1, 2, and 3 in our paper verify that our Transformer-MGKs/MLKs with 1 head, 2 heads, and 4 heads outperform or at least on par with the baseline softmax and linear transformers with double the number of heads, i.e., 2 heads, 4 heads, and 8 heads, respectively, on LRA and Wikitext-103. It is interesting to notice from Table 1 and 2 that even our 2-head Transformer-MGK/MLK models achieve better or equivalent results to the 12-head and 8-head baselines on the LRA benchmark. Using only half the number of heads or even fewer than that, our Transformer-MGKs/MLKs need much fewer parameters and FLOPs to compute than the baseline models. We believe this advantage of our models is significant.

On Point (2), Tables 1, 2, and 3 confirm that our Transformer-MGKs/MLKs with the same number of heads as the baseline models outperform the baselines. In particular, on the task that requires very long-range attention such as the retrieval task, our Transformer-MGKs/MLKs significantly outperform the baselines.  Note that this retrieval task has the longest average sequence length and attention span among the LRA tasks (Tay, Yi, et al. Long range arena: A benchmark for efficient transformers. ICLR 2021). Also, on the Wikitext-103 task, our Transformer-MGKs/MLKs with 4 heads significantly outperform the baselines with 4 heads. The advantage gaps between our Transformer-MGKs/MLKs with 8 heads and the baselines with 8 heads on the Wikitext-103 task are smaller because the baselines with 8 heads, given their capacity, already perform very well on this Wikitext-103 task. That says, the improvement in accuracy gained by Transformer-MGKs/MLKs in this case is indeed relevant.

**Theoretically, we have included a theorem in Section 2.2 in the revised manuscript** to show that our Transformer-MGKs/MLKs using a finite Gaussian mixture model can approximate sufficiently well any distribution of the queries at each time step, i.e., $P(q_i|t_{j}=1)$. This is not possible if only a Gaussian distribution is used as in the baseline softmax transformers. Thus, the GMM used in Transformer-MGKs/MLKs at each time step can capture a wider range of the distribution of the queries.

-----

We are glad to answer any further questions you have on our submission.

---

> ### Author Response · Authors · 2021-11-21
> **General Response - More on Empirical Evidences Showing the Advantages of Our Transformer MGK/MLK**
>
> Regarding the concern from the reviewers that there is not sufficient evidence showing an improvement in performance with our proposed Transformer with Mixture of Gaussian Keys model (Transformer-MGK), in addition to Point (1) and (2) in the previous comment,  we have run additional experiments to further verify that:
>
> (3) **The advantages of our Transformer-MGK/MLK models can be scaled up when applied on top of a larger and stronger baseline.**
>
> (4) **The advantages of our Transformer-MGK/MLK models still hold for different tasks including machine translation rather than just the LRA benchmark and Wikitext-103 language modeling.**
>
> Note that in all our experiments, we set **$M = 2$** for Transformer-MGK/MLK, i.e., using only 2 keys at each timestep.
>
> On Point (3), in Table 3 in our revised manuscript, we have provided additional empirical results on WikiText-103 when applied our MGK/MLK approach on top of a larger and stronger baseline adapted from the medium model in [1]. This model has 90M parameters, 16 layers, 8 attention heads per layer, and a hidden size of 256. **The size of our baseline model is close to BERTBase** [2], which has 110M parameters, 12 layers, 12 attention heads per layer, and a hidden size of 768. Note that the baseline transformer we used is deeper than BERTBase. This baseline transformer attains a test perplexity (PPL) of 29.60 as reported in [1], which, on this WikiText-103 task, is better than or equivalent to popular transformer models including [3], [4], [5], [6], and [7]. Note that for [7], results on WikiText-103 can be found in Table 6 of [8]. **Applying our MGK on top of this new larger and stronger baseline, we significantly improve the test PPL to 28.86**.
>
> On Point (4), we have studied the advantage of the MGK over the baseline transformer **on the IWSLT14 German-English machine translation task. Compared to the baseline transformer model that uses 4-head softmax attention, our models that use MGK with only 2 heads achieve better or comparable BLEU scores (34.69 and 34.34 vs. 34.42 BLUE scores of the baseline)**. We have summarized our results in Section 3.3 in the main text and Table 9 in Appendix A.10 of the revised manuscript. We provide details on the dataset, model, and training in Appendix A.1.3. Again, using only 2 heads vs. 4 heads as in the baseline allows our model to save more parameters and achieve better computational efficiency.
>
> **References**
>
> [1] Imanol Schlag, Kazuki Irie, and Jurgen Schmidhuber. Linear Transformers are Secretly Fast Weight Programmers. ICML, 2021.
>
> [2] Jacob Devlin, Ming-Wei Chang, Kenton Lee, and Kristina Toutanova. BERT: Pre-training of deep bidirectional transformers for language understanding. NAACL, 2019.
>
> [3] Edouard Grave, Armand Joulin, and Nicolas Usunier. Improving neural language models with a continuous cache. ICLR, 2017.
>
> [4] Yann N. Dauphin, Angela Fan, Michael Auli, and David Grangier. Language modeling with gated convolutional networks. ICML, 2017.
>
> [5] Stephen Merity, Nitish Shirish Keskar, and Richard Socher. An analysis of neural language modeling at multiple scales. arXiv, abs/1803.08240, 2018.
>
> [6] Jack W. Rae, Chris Dyer, Peter Dayan, and Timothy P. Lillicrap. Fast parametric learning with activation memorization. ICML, 2018.
>
> [7] Peter Shaw, Jakob Uszkoreit, and Ashish Vaswani. Self-attention with relative position representations. NAACL, 2018.
>
> [8] Zihang Dai, Zhilin Yang, Yiming Yang, Jaime Carbonell, Quoc V Le, and Ruslan Salakhutdi- nov. Transformer-xl: Attentive language models beyond a fixed-length context. ACL, 2019.

---

### Author Response · Authors · 2021-11-15
**Summary of the Revision**

Incorporating the comments and suggestions from all reviewers, besides fixing typos and notations, we have made the following main changes in the revised paper.

1. We have included a theorem in Section 2.2 in the revised manuscript to show that our Transformer-MGKs/MLKs using a finite Gaussian mixture model can approximate any distribution of the queries at each time step. The proof of this theorem is provided in Appendix C.


2. In all experiments, we set $M = 2$. Then we compare the baseline softmax and linear transformers with our Transformer-MGKs/MLKs ($M=2$) that use half the number of heads and the full number of heads. We’ve made this clear in the second paragraph of section 3 in the revised manuscript.


3. We have included a comparison of model complexity and computational cost between the Transformer-MGK and the baseline softmax transformer in Appendix B.


4. We have included results for softmax and linear transformers with 4 heads on WikiText-103 in Table 3 in the revised manuscript. For these baseline models, we see that reducing the number of heads from 8 to 4 significantly decreases the performance of the models with more than 1.5  reduction in validation and test PPL for the softmax transformer and more than 1.0 reduction in validation and test PPL for the linear transformer. Our proposed Transformer-MGK and Transformer-MLK help close this gap. We have made these points clearer in the Results paragraph in Section 3.2 of the revised manuscript.


5.  In Appendix A.4 of our revised manuscript, we have included an ablation study on the impact of the following factors on the performance of the Transformer-MGK: 1) the mixture of keys, 2) the Gaussian distance, and 3) the key shifting. The results are summarized in Table 5.


6. We have included a time overhead (wall-clock time) and GPU memory footprint analysis for our Transformer-MGK/MLK in comparison with the baseline softmax and linear transformers in Appendix A.5. We summarize the results in Table 6.


7. We have included a version of Figures 1 and 4 without log-scale and zoom in Appendix A.6 and showed that the results reported in those figures (Figures 9 and 10) converge.


8. We have included analysis of the learned  $\pi_{jr}$, $k_{jr}$, and $W_{K_{r}}$, which is the weight matrix projecting to $k_{jr}$, in Appendix A.7. The results are summarized in Figure 11, Figure 12, and Table 7.


9. We have provided additional analysis of computational complexity per training iteration of our Transformer-MGK/MLK models, the baseline softmax transformer, and the linear transformer in Appendix A.8. The results are summarized in Figure 13. Furthermore, Figure 14 shows FLOPs per training iteration of Transformer-MGKs that use different inference and learning methods.


10. We have updated Figure 3 (Upper Right, for FLOPs analysis) in the main text to reflect our latest results, which still show a significant advantage of our model over the baseline in terms of FLOPs.


11. We have updated “Learning via Stochastic Gradient Descent (SGD)” in section 2.3 to make it more concise and easier to read.


12. We have also corrected the typos in the “Design Options for Keys” paragraph in section 2.3 and the 2nd paragraph of section 3 to show that the Transformer-MGK/MLK  only needs $M$ projection matrices to compute the keys in each head if using option (A) to design the keys and only needs one projection matrix to compute the keys in each head if using option (B), i.e. with key shifting.

---

> ### Author Response · Authors · 2021-11-21
> **Summary of the Revision - More Updates in the Revised Manuscript**
>
> In addition to the changes in the previous comment, we have added the following updates into the revised manuscript.
>
> 13. In Table 3 in our revised manuscript, we have provided additional empirical results on WikiText-103 when applied our MGK/MLK approach on top of a larger and stronger baseline adapted from the medium model in [1].
>
> 14. In Section 3.3 in the main text and Table 9 in Appendix A.10 of the revised manuscript, we have provided additional empirical results to show the advantages of the MGK over the baseline transformer on the IWSLT14 German-English machine translation task.
>
> 15. We have added results for 12-head  softmax/linear transformers (vanilla) trained on the LRA benchmark into Tables 1 and 2, respectively, in our revised manuscript. We also provided additional results for 6-head Transformer-MGK and Transformer-MLK for the retrieval task in comparison with the 12-head softmax and linear transformers in Table 8 in Appendix A.9 of the revised manuscript.
>
> 16. We have updated our Related Work section to include a comparison of Transformer-MGK/MLK models with [2] and [3].
>
> 17. We have added a comparison between our Transformer-MGK and the multi-query attention [4] in Appendix A.11 and cited [4] and [5] in the Related Work section of our revised manuscript.
>
> **References**
>
> [1] Imanol Schlag, Kazuki Irie, and Jurgen Schmidhuber. Linear Transformers are Secretly Fast Weight Programmers. ICML, 2021.
>
> [2] Rami Al-Rfou, DK Choe, Noah Constant, Mandy Guo, and Llion Jones. Character-level language modeling with deeper self-attention. AAAI, 2019.
>
> [3] Alexei Baevski and Michael Auli. Adaptive input representations for neural language modeling. ICLR, 2019.
>
> [4] Noam Shazeer. Fast transformer decoding: One write-head is all you need. arXiv preprint arXiv:1911.02150, 2019.
>
> [5] Hao Peng, Nikolaos Pappas, Dani Yogatama, Roy Schwartz, Noah Smith, and Lingpeng Kong. Random feature attention. ICLR, 2021.

---

### Author Response · Authors · 2021-11-21
**Any Questions before the Deadline to Update Our Draft?**

Dear reviewers,

We would like to thank all reviewers again for your thoughtful reviews and valuable feedback. We have just updated our manuscript and added new replies to your comments and questions with our latest experimental results. We have summarized the changes we made in the manuscript in the Summary of the Revision below.

We would appreciate it if you could let us know if there are additional questions or concerns before the deadline to update the draft.

We would be happy to do any follow-up discussion or address any additional comments.

---

### Decision · Program_Chairs · 2022-01-20

**Decision:**

Reject

**Comment:**

The paper proposes using a mixture of Gaussian models for transformer keys (MGK) so that the posterior distribution of key given query matches the attention scores in the transformer architecture under some assumption. Similarly but in reverse, the query given key under a MoG also matches transformer attention score. The paper proposes that this formulation can learn more diverse attention-heads and reduce the computation by replacing some heads with the Gaussian mixture heads which are easier to compute. Moreover, the authors show that it is straightforward to transfer their formulation to linear transformers. The authors perform experiments on the Long Range Arena benchmark and on Wikitext-103 where they show some improvements with less attention heads, while using up to 20% less FLOPs for softmax transformers and around mostly 20% but up to 80% less parameters for the worse linear transformer.

Reviewers generally find that the interpretation of transformers as mixture of Gaussian interesting, although it is based on normalization assumptions. The main objections from both positive and negative reviewers is the weakness of the empirical results. In summary, all the improvements in perplexity are less than 0.5, and accuracy improvements are less than 1. In the discussions, the authors claim that their comparisons are using fewer heads vs baselines with more heads, which is how the tables are presented, but upon closer look I find this unconvincing since the same head count comparisons can be reconstructed and still show very weak results. For example in table 1, acc(MGK4)=61.85 vs. acc(baseline4)=61.23, or in table 2 acc(LinearMG4)=55.7 vs acc(LK4-baseline)=55.61. To begin with, it would be better to compare the same number of heads for more head-to-head comparisons, so I find the whole argument to be misleading. The claims about FLOPS and memory has a similar flavor where the differences are small but the claims were big. A visual evaluation of figure 3 show that most reductions are around 10% in FLOPs or parameters for the better softmax model (note the axis does not start at 0). The retrieval task for the worse linear transformer is the only case where the FLOPs reduction seems significant. Given that EM is now required during training, I'd interpret the results as a negative for MGK given the marginal improvements achieved.

Several reviewers are unhappy with the strength of the empirical results while most reviewers gave favorable scores after the discussion where the authors insisted that some valid points are not valid. The authors requested the AC to look further into the one negative review due to the lack of a reviewer's response. After taking some time to look at the substance of the paper and the review, I recommend rejection due to the weakness of the results, the misleading presentation and discussions.